# Global total precipitable water variations and trends during 1958-2021

Nenghan Wan[1], Xiaomao Lin[1*], Roger A. Pielke Sr.[2], Xubin Zeng[3], and Amanda M. Nelson[4]

[1]Department of Agronomy, Kansas Climate Center, Kansas State University, Manhattan, KS.
[2]Cooperative Institute for Research in Environmental Sciences, University of Colorado, Boulder, CO.
[3]Climate Dynamics and Hydrometeorology Center at the University of Arizona, Tucson, AZ.
[4]National Center of Alluvial Aquifer Research, USDA-ARS Sustainable Water Management Research Unit, Stoneville, MS.

*Correspondence to*: Xiaomao Lin (xlin@ksu.edu)

**Abstract.** Global responses of the hydrological cycle to climate change have been widely studied but uncertainties still remain regarding water vapor responses to lower tropospheric temperature. Here, we investigate the trends in global total precipitable water (TPW) and surface temperature from 1958 to 2021 using ERA5 and JRA-55 reanalysis datasets and further validate these trends by using radiosonde from 1979 to 2019, Atmospheric Infrared Sounder (AIRS) and Microwave Satellite (SSMI(S)) observations from 2003 to 2021. Our results indicate a global increase in total precipitable water (TPW) of ~2 % per decade from 1993-2021. These variations in TPW reflect the interactions of global warming feedback mechanisms across different spatial scales. Our results also revealed a significant near-surface temperature ($T_{2m}$) warming trend ~0.15 K per decade during 1958-2021. Its consistent warming at a rate of ~0.21 K per decade after 1993 corresponds to a strong water vapor response to temperature at a rate of 9.5 % $K^{-1}$ globally, with land areas warming approximately twice as fast as the oceans. The relationship between TPW and $T_{2m}$ showed a variation around 6 - 8% $K^{-1}$ in the 15-55 $^o$N latitude band, aligning with theoretical estimates from the Clausius–Clapeyron equation.

## 1 Introduction

As the most critical greenhouse gas in the Earth's atmosphere, water vapor plays a key role in atmospheric processes from the microscale (including the formation of clouds and precipitation) to the global scale, and is related to the Earth's radiation budget, hydrological cycle, and climate change (Held and Soden, 2006; Lacis et al., 2010; Ruckstuhl et al., 2007). The amount of water vapor is primarily controlled by the air temperature when the relative humidity (especially over the ocean) remains unchanged in the low troposphere. The total precipitable water (TPW), also known as the column-integrated amount of water vapor from the surface to the top of the atmosphere, increases by 6-7% with a 1 K increase in air temperature according to the Clausius-Clapeyron equation and thus enhances the strength of global warming with strong positive feedback due to the greenhouse effect (Held and Soden, 2006, O'Gorman and Muller, 2010) particularly in upper troposphere whereas changes at lower levels are strongly linked with precipitation patterns, influencing the frequency and intensity of extreme weather events (Trenberth, 1998; Trenberth et al., 2003). Therefore, evaluating the long-term trend of TPW change and its relationship with

temperature is important for understanding the role of water vapor in climate change and the impact of water vapor feedback on global warming.

Because of the short residence time of water vapor in the atmosphere, studies in terms of long-term water vapor trends and their variability still face challenges. The two main categories of water vapor data include observations collected from various weather station networks and satellites, as well as reanalysis datasets integrated with simulations and observations. The former has difficulty in evaluating long-term trends due to discontinuities, insufficient spatial densities, and coverage (Dee et al., 2011). For example, radiosonde observations have the issue of spurious discontinuities or changes in variability due to shifts in instruments and observational methods (Dee et al., 2011; Trenberth et al., 2005). However, multiple observations have been homogenized and thoroughly quality-checked in recent years. For example, the ground-based measurements of Global Navigation Satellite Systems (GNSS) integrated water vapor from 1994-2021 (Bock, 2022) and provided evidence of global moistening (Douville et al., 2022). The in-situ observations from the Radiosounding HARMonization (RHARM) dataset (Madonna et al., 2022) enhance the spatial consistency of estimated trends and align more closely with a contemporary atmospheric reanalysis. In addition, the homogenized temperature data presented by Zhou et al. (2021) exhibited spatially consistent trends and temporally stable variations, and didn't show the erroneous tropospheric cooling observed in various reanalyses including ERA5 and raw datasets, across North China and Mongolia. This accuracy of temperature can help in increasing the confidence in tropospheric temperature and water vapor, as well as in enhancing the quality of atmospheric reanalysis products. Temperature and humidity extremes from HadISDH by Willett (2023) are also designed for long-term regional trends. The latter reanalysis dataset has issues of data quality that suffered from biases and errors from assimilated data during the satellite era, causing concerns about their reliability for detecting climate trends (Dai et al., 2011; Schröder et al., 2016; Trenberth et al., 2011). Trenberth et al. (2005) evaluated the performance of global reanalyses, satellite, and radiosonde datasets on TPW and found that the Special Sensor Microwave Imager Sounder (SSM/I) dataset provided by Remote Sensing Systems (RSS) is the only source with reliable means, variabilities, and trends for TPW over oceans after 1988. The discontinuity and inaccurate data in 1992 due to changes in satellite instruments was highlighted by Trenberth et al (2015) and this spurious variability in water vapor existed in reanalyses up to the present. Numerous studies have analysed trends and variations in atmospheric water vapor distribution on both global and regional scales, primarily utilizing early versions of reanalysis datasets and satellite observations, albeit over relatively short study periods (e.g., Borger et al., 2022; Parracho et al., 2018; Wang et al., 2016; Zhang et al., 2019, 2021). New generation reanalysis datasets (JRA-55 and ERA5) provide a better option for climate studies because of advanced modeling and data assimilation systems, with better accuracy, and fewer homogeneity issues (Hersbach et al., 2020; Kobayashi et al., 2015; Douville et al., 2022). JRA-55 showed a better performance in studying multidecadal variability and climate change than previous reanalysis datasets (Kobayashi et al., 2015). Many studies have confirmed that ERA5 was the best or among the highest-performing reanalysis products (Taszarek et al., 2021; Yuan et al., 2023) although inhomogeneity still remains; for example, water vapor associated with changes in SSMI instruments (Trenberth et al., 2015) and unreliability of tropical water vapor in ERA5 and ground-based observations before

1993 (Allan et al., 2022). Therefore, the latest ERA5 and JRA-55 were selected in this study to analyze long-term TPW changes and their relationship with temperatures at regional and global scales.

The near-surface air temperature (2m air temperature, $T_{2m}$) describes the thermodynamic temperature at a 1.5-2 m height while surface skin temperature ($T_s$) refers to 'radiometric surface temperature' that is governed by the terrestrial radiation balance
(Jin et al., 1997; Jin and Dickinson, 2010). Recent studies investigated how the TPW responds to changes in surface temperature using modeling estimation, satellite observations, or ground-based observation systems (Allan et al., 2022; Alshawaf et al., 2017; Borger et al., 2022; O'Gorman and Muller, 2010; Wang et al., 2016; Yuan et al., 2021), and many global surface temperature observational datasets use sea surface temperature (SST) over ocean and $T_{2m}$ over land (e.g., HadCRUT, Morice et al., 2021). Different from prior studies, we use the newest and longest available reanalysis datasets to discuss the
difference of $T_{2m}$ and $T_s$, and the relationship between trends in TPW and $T_{2m}$ from 1993 to 2021 along with additional analysis using radiosonde data as well as AIRS and SSMI(S) satellite measurements since 2003.

In this study, we focus on answering the following questions: (1) To what extent has the TPW changed and what is the difference between the variations of $T_{2m}$ and $T_s$ on a multi-decade scale? (2) What is the relationship between trends of TPW
changes and temperature changes? The results from reanalysis dataset are compared with radiosonde data as well as AIRS and SSMI(S) satellite measurements. We discuss the difference of results among datasets and their potential discontinuities in datasets. Therefore, this paper is organized as follows. In section 2, we introduce the datasets and methods. In section 3, the TPW variations and the differences of $T_{2m}$ and $T_s$ are compared with radiosonde and satellite observations over land and ocean. Sections 4 and 5 provide the discussion and conclusion.

**2 Data and Methods**

**2.1 Datasets**

Two reanalysis datasets containing TPW and temperature ($T_{2m}$, $T_s$) from 1958 to 2021 were used in this study, these being ERA5 (Hersbach et al., 2020) and JRA-55 (Kobayashi et al., 2015). For ERA5, skin temperature, $T_s$, is the temperature estimated from the surface energy balance. For JRA-55, skin temperature is a diagnostic variable computed from the surface
upward longwave radiation under the assumption that the surface is a black body. We used monthly reanalysis datasets from January 1958 to December 2021 for TPW and temperatures. Both TPW and temperature variables in these two datasets were regirded into 1° x 1° resolution data using a bilinear interpolation method before analyzing (Zhuang, 2018).

The Atmospheric Infrared Sounder (AIRS) instrument captures a precisely calibrated, spectrally detailed dataset of both
infrared and microwave radiances. It offers quality temperature and humidity profiles throughout the troposphere (Tian et al., 2020). Satellite observation from version 7 of AIRS (Tian et al., 2020) was used for comparison with the reanalysis from 2003

to 2021. The total column water vapor (kg m$^{-2}$) is calculated as the average of daytime (TotH2OVap_A) and nighttime modes (TotH2OVap_D). In addition, the column-integrated water vapor from the Special Sensor Microwave Imager and the Special Sensor Microwave Imager Sounder (SSMI(S)) from 2003-2021 (Wentz et al., 2015) are selected for comparison with reanalysis datasets.

We also used *in-situ* observations from the Radiosounding HARMonization (RHARM) dataset (Madonna et al., 2022), which applied the new algorithm to the Global Climate Observing System Reference Upper-Air Network (GRUAN) data and used observation measurement instead of reanalysis data as a reference to calculate and adjust for systematic effects on temperature and humidity. RHARM provides homogenized temperature and relative humidity profiles at two observation times (0000 and 1200 UTC) for radiosonde stations globally from 1979 to 2019. Therefore, $T_{2m}$ used in this study from radiosonde observation was taken from the temperature observed at the lowest layer. Relative humidity and temperature from the surface to 500 hPa level are used to calculate precipitable water.

## 2.2 Methods

Monthly TPW and temperature anomalies were calculated by the base period of 1958-2021 for reanalysis datasets and discussed the trends of long-term (1958-2021) and short-term (1993-2021) periods. We divided the globe into tropical regions (23.5°S–23.5°N), temperate (23.5°S–66.5°S in Southern Hemisphere (SH) and 23.5°N–66.5°N in Northern Hemisphere (NH)), and polar regions (66.5°S–90°S in SH and 66.5°N–90°N in NH). The regional average values, presented by different latitude bins, were calculated with a cosine (latitude) weighting factor to account for the convergence of grid points for each region. For the global distribution, all datasets were re-sampled into 1° x 1° resolution using spatial-averaging resampling, then the area-weighted average of anomalies was computed to formulate a global time series.

For all trend analyses in TPW and temperature ($T_s$, $T_{2m}$) series, we selected the Seasonal Kendall (SK) test (Hirsch et al., 1982) using the Theil-Sen slopes (Sen 1968; Theil, 1992) to calculate relative TPW trends (% dec$^{-1}$) (Zhai and Eskridge, 1997) and temperature trends for both $T_s$ and $T_{2m}$ (K dec$^{-1}$). The statistical significance of all linear correlations and trends used was performed at a 95% confidence level for all analyses conducted in this study. The TPW responses to temperature from 1993 to 2021 is calculated as the ratio of TWP trends and $T_{2m}$ trends.

Due to daily data missing in radiosonde observations, valid data required at least 10 days of data available within a month, and at least two-thirds of the total months had to have valid monthly data (345 months for 1979-2019). Each month had to have at least 28 years of valid data at each station to calculate monthly climatology. Trend analysis is also from Sen's estimate of the slope (Sen 1968) for these observations. Thereby, a set of 331 radiosonde stations were selected in this study. To facilitate the comparison, interpolated reanalysis data from radiosonde stations were selected to compare with the observation data. We

calculated the trend difference between reanalyses and observations to assess how effectively each reanalysis product captures

changes in observed TPW over 1979-2019.

To detect potential discontinuity within temporal sequence of total water vapor and temperature in reanalysis dataset, we applied penalized maximal F test (PMF test, Wang et al., 2006, 2013; Wang, 2018) to detect temporal discontinuity points on each grid over ocean during 1958-2021. The discontinuities were documented if testing statistic's p-value was less than 0.01

(Zhou et al., 2021).

## 3 Results and Discussion

### 3.1 TPW Trends

Figures. 1a-e show the temporal variations of monthly TPW anomalies over tropical, temperate, and polar regions. Overall, in terms of latitude bands, these trends differ little between the two reanalysis datasets. Specifically, ERA5 gives a 0.6 % dec$^{-1}$

moistening rate whereas the JRA-55 shows a 0.96 % dec$^{-1}$ moistening rate over the tropical regions for the whole period 1958-2021. A notable decrease in TPW is shown during the 1980s to 1990s over tropical regions in ERA5, which is consistent with ERA-interim (Allan et al., 2014) but inconsistent with satellite microwave data (Allan et al., 2022). This discrepancy seems to originate in the tropical lower troposphere over ocean. Since SSM/I satellite became available and were used in ERA5 after 1987 (Hersbach et. al., 2020), there has been some spurious variability in TPW, notably in January 1992 (Trenberth et al.,

2015). However, a strong agreement of TPW after 1993 is shown between SSM/I observations and ERA5 (Allan et al., 2022). In addition, data after 1993 show fewer change points based on PMF-test results on grid boxes over the ocean (Figure S2). Thus, the 1993-2021 time series of TPW benefiting from in-situ and satellite observations is expected to produce more reliable water vapor variability. ERA5 gives a 1.46 % dec$^{-1}$ moistening rate whereas the JRA-55 shows a 1.43 % dec$^{-1}$ moistening rate over the tropical regions from 1993 to 2021. Trends range from 1.64 to 3.89 % dec$^{-1}$ over temperate and polar regions in NH

and are about 1.56 (0.89) % dec$^{-1}$ over temperate but are not significant over polar in SH for ERA5 (JRA-55) during 1993-2021. The TPW trends increase in the NH is larger than those in the SH although they showed strong month-to-month variabilities (coefficient of variation, CV, is 0.7 on average) in the NH, which are consistent with the global distribution (Fig. 1).

Figures 1f-h depict these trends of TPW estimated over the ocean, land, and the globe. It is noticeable that monthly TPW anomalies showed a consistent and statistically significant increase, yielding a rate of 0.58 (0.85), 0.88 (1.07), and 0.65 (0.89) % dec$^{-1}$ for ERA5 (JRA-55) during 1958-2021 over ocean, land, and the globe, respectively. However, those trends increase at a rate of 1.72 (1.21), 1.19 (2.22) and 2.12 (1.91) % dec$^{-1}$ for ERA5 (JRA-55) after 1993. The monthly TPW anomalies showed a similar variability over ocean, land, and globally (CVs, around 0.4), and displayed significant interannual variations,

which had been dominated by ENSO events (Trenberth et al., 2005; Wang et al., 2016). The distinct turning points may be

attributed to the intensity of ENSO events. For example, one of the most powerful ENSO events during the 1997/1998 El Niña led to a significant tropical TPW increase, attributable to the warming in the equatorial Pacific (Wagner et al. 2005). The 2015/2016 El Niño caused a moistening TPW trend over tropical regions (Garfinkel et al., 2018). The trend of TPW during 1988-2003 was likely associated with the decadal variation of Interdecadal Pacific Oscillation from a warm period (1977-1998) to a cold period (1999 to 2003) (Dong and Dai, 2015). The TPW trend for recent period including 1997/98 event will likely be similar (Wang et al. 2016). In addition, Patel and Kuttippurath (2023) also shows a strong correlation (0.81) between Pacific Decadal Oscillation and TPW variability in the tropics.

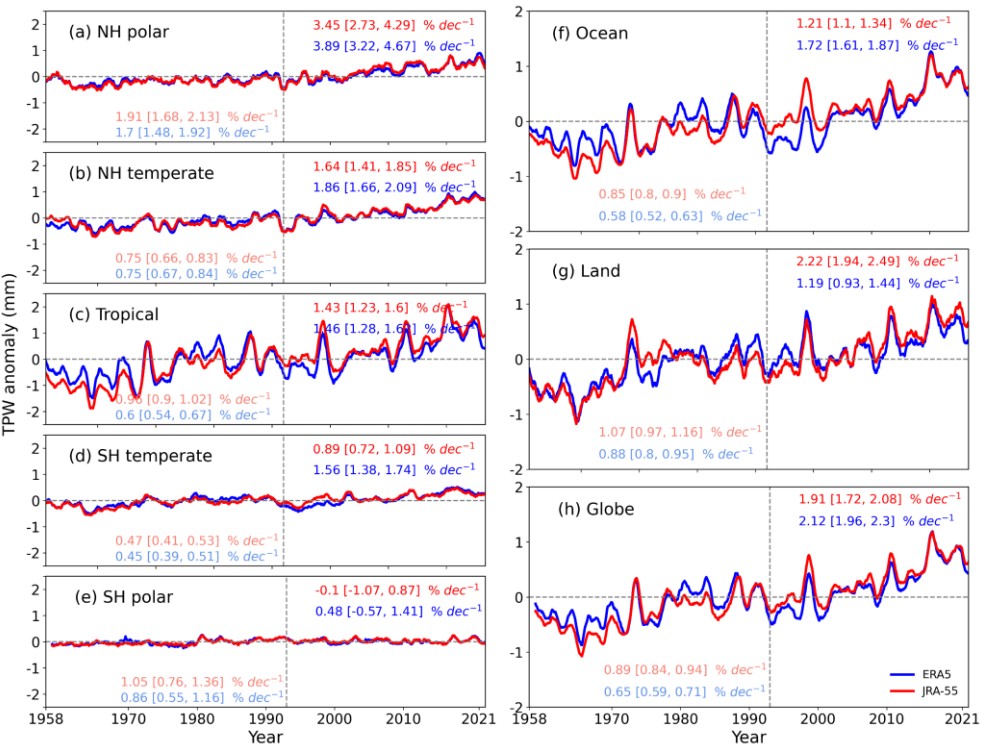

**Figure 1.** Time series of monthly TPW anomalies (mm) over **(a)** the Northern Hemisphere (NH) polar, **(b)** NH temperate, **(c)** Tropical, **(d)** Southern Hemisphere **(SH)** temperate, **(e)** SH polar, **(f)** ocean area, **(g)** land area, and **(h)** the globe from the ERA5 (blue) and JRA-55 (red) during the period 1958–2021. A 12-month running smoother was applied to each time series shown. The trends are shown for 1958-2021 (light color) and 1993-2021 (dark color) periods with 95% confidence intervals in brackets. The vertical line indicates the year of 1993.

Based on timeseries of TPW in Fig.1, Fig. 2 shows the decadal trends in TPW calculated from ERA5 and JRA-55 after 1993. In general, the two reanalysis datasets showed similar trend patterns of global TPW distribution, with upward trends as the dominant change of TPW, indicating a rise in moisture in response to global warming since the end of the twentieth century (Santer et al., 2006). TPW trends are largely positive and statistically significant over North Africa, Europe and the NH polar

region. In addition, Simpson et al. (2023) found that near-surface water vapor, as measured by surface observations, hasn't
increased over arid and semi-arid regions since 1980, a discrepancy compared to predicted results from simulations. These
findings indicate there might be misrepresentations of hydroclimate-related processes in simulations since climate models
showed moistening trends associated with the increase in water vapor-holding capacity of a warmer atmosphere. On the other
hand, decreasing TPW trends show a dipole structure over the Southern Ocean in both reanalysis datasets which can be
attributed to the change of the ENSO phase (Trenberth et al., 2005). In contrast, these two reanalysis datasets show opposite
trends over eastern Africa including the Sudan where ERA5 trends are negative, whereas JRA-55 trends are positive. The
reason for opposite trends in reanalyses is likely due to the different representations of large-scale moisture transport, surface-
atmosphere processes, and their data assimilation (Parracho et al., 2018). Chen and Liu (2016) and Parracho et al. (2018) also
showed decreasing trends in North Africa using ERA-Interim reanalysis for 1979-2014 and 1980-2016, and the biases in
rainfall over West Africa might also exist in ERA-Interim reanalysis (Dunning et al., 2016). In terms of TPW trend over arid
and semi-arid regions from 1980 to 2020 in reanalysis dataset (figure not shown), it also shows a different trend, comparing
observation from Simpson et al. 2023, that only TPW over southwest United States in ERA5 and central Africa in JRA-55
significantly decreased.

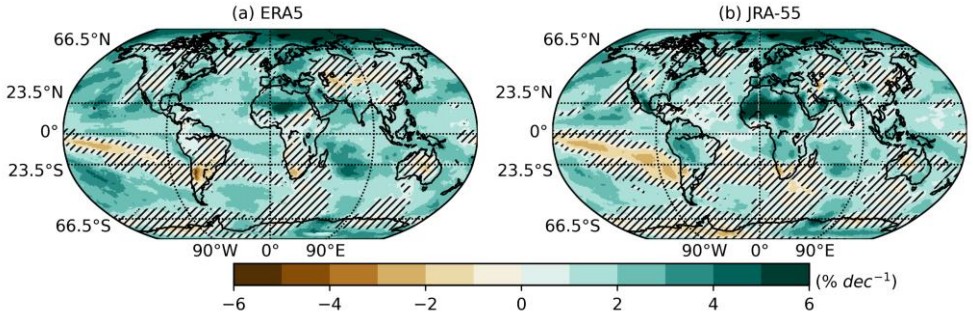

**Figure 2.** Monthly total precipitable water (TPW) trend (% dec$^{-1}$) from 1993 to 2021 from **(a)** ERA5 and **(b)** JRA-55. The
hatch areas represent trends that are not statistically significant at a 95% confidence level.

### 3.2 Trends in temperature

In contrast to TPW, temperature in ERA5 is relatively stable for the pre-satellite era and aligned with independent datasets,
although some notable discrepancies exist in accurately representing specific regional variations (Hersbach et al. 2020). The
global trends of temperature between 1958-2021 exhibit widespread warming over large regions in both reanalysis datasets,
except for some of small areas located in the SH and northern Atlantic (Fig. 3). Large land areas experience warming
temperature changes of greater magnitude compared to the surrounding oceans (Fig. 3) and the strongest warming occurred in
the Arctic (Fig. 3). Greater warming is observed over mid-latitude than tropical regions in NH, consistent with Zeng et al.
(2021). Cooling in the North Atlantic is also observed by Li et al. (2022) when using observation datasets. This arises from
strengthened local convection and heat dissipation from the ocean induced by the overlying atmosphere. Meanwhile, the North

Atlantic cooling appears most significant in February in the northern hemisphere while it becomes net warming in August (Allan and Allan, 2020). Some studies connect this phenomenon to changes in ocean circulation based on modeling results (Drijfhout et al., 2012). In addition, based on model simulations without considering variable ocean currents, He et al. (2022) explained that the warming hole is driven by enhanced surface westerly wind removing heat from ocean surface. For the SH, the polar region also experiences the strongest warming trends, similar to the trend's magnitudes by Clem et al. (2020). There is a slight difference between two reanalysis datasets for differential trends ($\Delta T$) between $T_s$ and $T_{2m}$ at the global scale, especially over the ocean area where $\Delta T$ trends are negative in ERA5 and positive in JRA-55. The $\Delta T$ shows a similar pattern only over the tropic land region (Fig. 3d). Although air and surface temperatures are closely related, they are physically distinct. The difference may be affected by changes of regional and seasonal vegetation ecosystems, land use, and land cover (Gulev et al., 2021; Masson-Delmotte, 2022). The ability of atmospheric reanalyses to effectively constrain variations in $T_s$ and $T_{2m}$ trends is limited because the sea surface temperatures in atmospheric reanalyses were obtained from globally observed and interpolated products (Rayner et al. 2003). In contrast, the estimation of air temperature depends on model parameterizations and assimilated observations, which do not incorporate marine air temperature (Simmons et al., 2017). The strong correlations between $T_{2m}$ and $T_s$ are further discussed in SI (Fig. S4 &S5). Overall, $T_{2m}$ and $T_s$ show very similar interannual variations and the trends aren't significantly different. Therefore, we only present $T_{2m}$ variations and its relationship with water vapor.

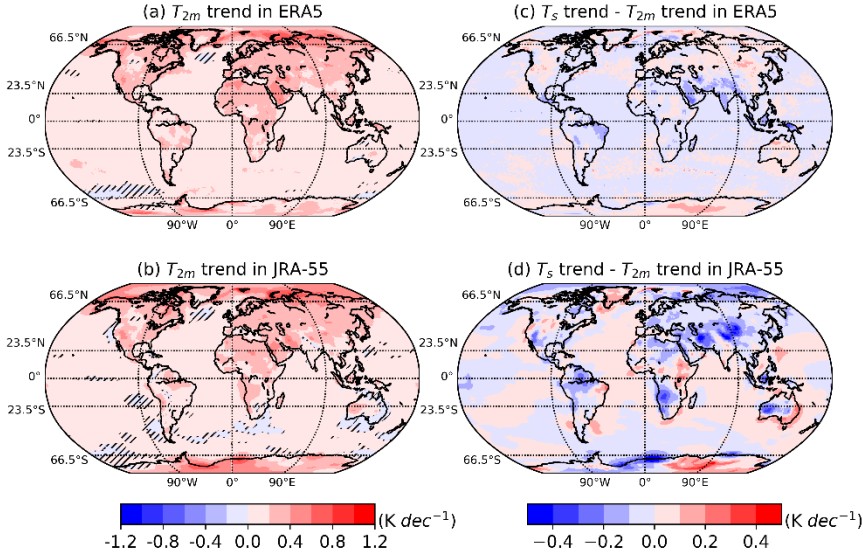

**Figure 3.** Trends of surface air temperature ($T_{2m}$) from 1958 to 2021 for ERA5 **(a)** and JRA-55 **(b)**. The differential trends between $T_s$ and $T_{2m}$ trends for ERA5 **(c)** and JRA-55 **(d).** The hatch areas indicate trends that are not statistically significant. Trend units: K dec$^{-1}$.

The general warming since the late 1970s revealed by $T_{2m}$ over different latitude bins in both datasets agree with each other when viewing the spatial distribution of magnitudes globally (Fig. 4), manifested by much larger warming in the Arctic at a

rate (all temperature rates given as K dec$^{-1}$) of ~0.45, which is three times warming than the global average rate (~0.15) (Figs. 4h). During the 1979-2021 period, this Arctic amplification of warming was markedly pronounced, occurring at rates 3.5 times faster than the global average warming rate documented by surface climate observations (Rantanen et al., 2022) in ERA5 data and 4 times faster in JRA-55 data. The potential causes of the Arctic amplification may be linked to sea ice decline, changes in atmospheric and oceanic heat contents, or changes in atmospheric moisture transport (Graversen, 2006; Screen and Simmonds, 2010; Zhang et al., 2008). The Arctic warming is concentrated in the lower troposphere (Allan et al. 2022) and reached ~0.74 K dec$^{-1}$ after 1993, thus leading to increases in water vapor as shown in Fig. 1a. Antarctic warming is at a rate of ~0.24 K dec$^{-1}$ over the past 64 years and consistent warming at ~0.23 K dec$^{-1}$ after 1993. In addition, the warmer and colder years are linked to the ENSO events and further cause the distinct turning points of TPW (Fig. 1). The datasets exhibit a clear consensus regarding the increased temperature of both land and ocean areas over the years. The land experienced greater warming at a rate exceeding 0.2 for $T_{2m}$ compared to the oceans' warming, leading to a land-to-ocean warming ratio ranging from 1.8 to 2.3 (Fig. 4). The explained reasons for the land-ocean contrast in warming rates is the influence of temperature and humidity on lapse rate, which results in a smaller reduction in lapse rate over land areas compared to the ocean, leading to a greater warming effect on land surfaces than on ocean surfaces (Joshi et al., 2008). Additionally, the restriction of moisture in the land boundary layer directly contributes to an intensified warming of land surfaces, which, in turn, raises the lapse rate over the land (Joshi et al., 2008). Byrne & O'Gorman (2018) further extended Joshi's theory in a quantitative way. They investigated the contrast of land/ocean warming from surface observations and model simulations at the global scale and indicated that amplified land temperature increases were the consequence of reduced relative humidity over land.

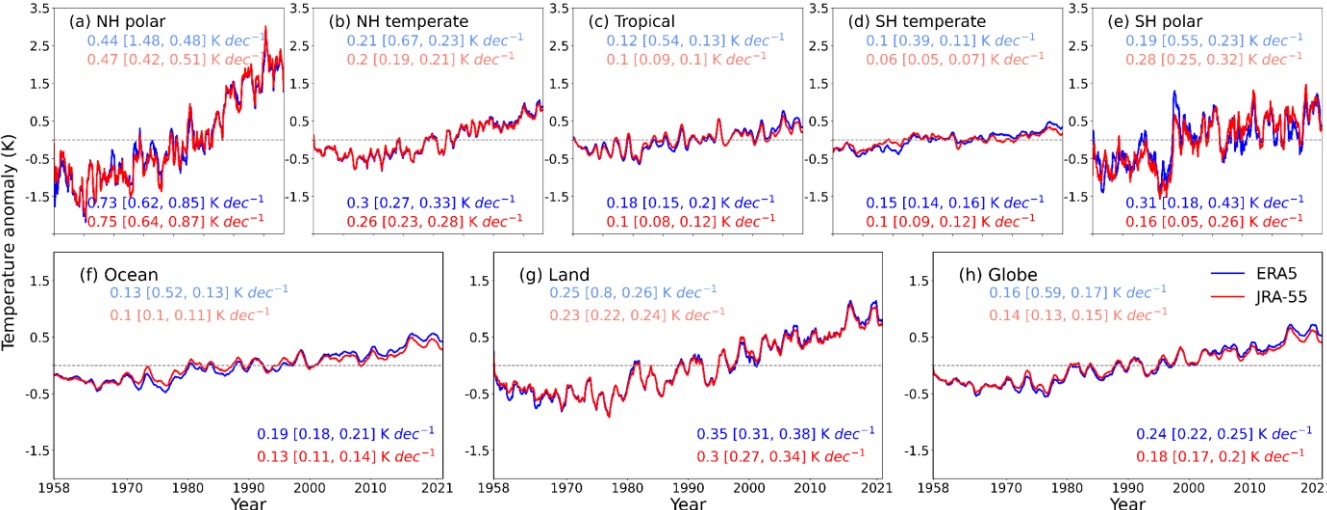

**Figure 4.** Monthly $T_{2m}$ and $T_s$ anomaly over **(a)** the Northern Hemisphere **(NH)** polar, **(b)** NH temperate, **(c)** Tropical, **(d)** Southern Hemisphere **(SH)** temperate, **(e)** SH polar, **(f)** ocean, **(g)** land, and **(h)** the globe from the ERA5 (blue) and JRA-55 (red) data sets during the period of 1958 to 2021. A 12-month running smoother was applied to all time series. The trends are

shown for 1958-2021 (light color) and 1993-2021 (dark color) periods with 95% confidence intervals in brackets. The vertical line indicates 1993 year.

### 3.3 TPW change response to temperature

TPW is expected to increase with air temperature by about 7% per K according to the Clausius–Clapeyron equation if the relative humidity in the lower troposphere is constant (Trenberth et al., 2005; O'Gorman and Muller, 2010). This relationship is determined by the ratio (dTPW/dT) of TPW trends and temperature trends from 1993 to 2021 (Fig. 5). The dTPW/dT ratio shows a similar pattern but different magnitude between two reanalysis datasets. The TPW increases significantly around 11.2% (10.9%), 5.8% (9.4%), and 9.5% (8.9%) per K for ocean, land, and globe for ERA5 (JRA-55). These rates are larger than 6% per K at a global scale based on observational HadCRUT5 (Allan et al., 2022). A disagreement involving TPW trend and tropospheric temperature trend between observation and modelling simulation is also shown in Santer et al. (2009, 2021). The dTPW/dT ratio increases significantly around 6% per K over eastern North America and Europe for both ERA5 and JRA-55 (Fig. 5). However, the relationship between TPW patterns and rising surface temperatures does not always follow the Clausius-Clapeyron equation, especially over some areas of Asia, central Africa, Australia and South American where TPW response to rising surface temperatures is negative without significance (Fig. 5). Regions with higher data density and homogeneity, such as North America and Europe, are likely to exhibit more realistic trends. Negative dTPW/dT occurs over the southern tropical ocean, where sea surface temperature was increased (Fig. 3) but precipitable water appears to be decreased slightly (Fig. 1); thus, the TPW change is contrary to what might be expected from the thermodynamic response to changes in temperature. Water vapor moves from one location to another through circulation, and the trend of vertical motion at 500 hPa could be further investigated to reveal the ascent and convective activity as well as its relationship to decreasing water vapor (Zveryaev and Allan, 2005).

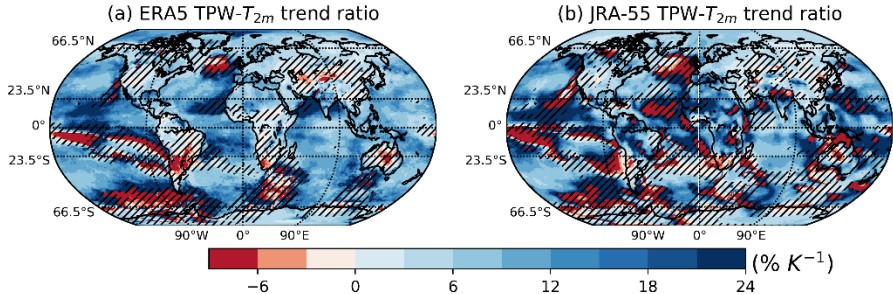

**Figure 5. (a)** The ratio (% K$^{-1}$) of the TPW trends and the surface air temperature (T$_{2m}$) trends from 1993 to 2021 in ERA5. **(b)** Same as **(a)** but for JRA-55. The hatch areas represent ratios that are not statistically significant at the 95% confidence level.

In addition to the global patterns of spatial dTPW/dT ratios, the ratio's latitude dependency is an interesting subject (Fig. 6) between 55°S and 55°N. Although the TPW response values show strong variations across latitudes, the ratios show a similar changing pattern over the globe and ocean, which varies across the theoretical Clausius-Clapeyron curves of dTPW/dT ratios (Figs. 6a). The dTPW/dT ratios are close to the Clausius-Clapeyron response curves in NH mid- to high-latitudes over the ocean, but larger than 6% per K in the SH (-40 to -20°N band) over the ocean. Such strong latitude dependency and the discrepancy between land and ocean areas are associated with zonal relative humidity changes and possible amplification of surface warming over land relative to the ocean (O'Gorman and Muller, 2010) (Figs. 6c). In arid areas, due to a lack of water, warming doesn't result in increased water vapor; conversely, in monsoon regions, the dynamics of the atmosphere amplify moisture increases (Fasullo, 2011). Multiple other studies also confirm the "dry gets drier, and wet gets wetter" paradigm over land (Xiong et al., 2022). Likewise, an analysis of the salinity index in Cheng et al. (2024) showed that salty areas are getting saltier and fresh areas are getting fresher; and that over the ocean wet areas are getting wetter and dry areas are getting drier. There are two stronger TPW response zones, located in the southern mid-latitude and the tropics over the globe (Figs. 6a, c). This result is nearly the same when drawn from multiple models of climate simulations for both historical and projected climate scenarios (O'Gorman and Muller, 2010). For land areas, in addition to two stronger response zones similar to the globe and ocean, a local maximum was found in the sub-tropical areas of the NH and SH (Fig. 6). The ratios for the land region are mainly lower than the theoretical ratio (around 7% per K) and these ratios are even negative in SH mid-latitudes where is the arid and semi-arid regions. Comparing these ratios between reanalyses, their discrepancies are greater over ocean, which probably contributes to their discrepancies over the globe (Fig. 6).

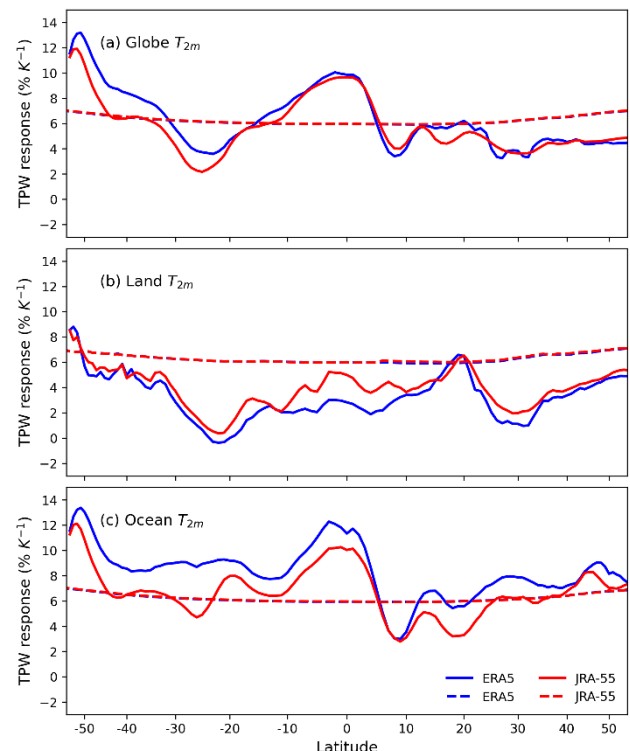

**Figure. 6**. The meridional means of change of TPW response to $T_{2m}$ for ERA5 (blue) and JRA-55 (red) over **(a)** globe, **(b)** land, and **(c)** ocean base using data from 1993 to 2021. The dashed lines represent the theoretically expected Clausius-Clapeyron response based on the climatological zonal mean temperature from the trend analysis.

## 4 Further Discussion

The reanalysis dataset combines various observations to create a coherent global dataset using an atmospheric general circulation model. It provides spatially coherent and consistent data with global coverage, which makes it valuable for climate research (e.g., Allan et al., 2022; Urraca and Gobron, 2023). By assimilating multiple data sources and employing a physics-based model, reanalyses can effectively address uncertainties and minimize the presence of unrealistic values. However, it is worth mentioning that discontinuities in the time series may still occur in reanalysis datasets when satellites and conventional observations are transited, newly added, and removed during the assimilation process (Long et al., 2017). In order to get a general picture of discontinuities in the reanalysis data, we calculated the discontinuity from 1958 to 2021 for each grid box of temperature and TPW over the ocean, where there is a strong relationship between water vapor and temperature along with significantly increased trends (Figs. 1 and 3). We applied the PMF test on each grid of the reanalysis dataset and selected the significance level of 0.01 to detect discontinuities (Zhou et al., 2021). The years with detected discontinuities (after counting all change points within the same year) of both temperature and TPW are shown in Figures S1 and S2. Discontinuities for

temperature are relatively frequent before 1980 in both reanalysis datasets (Fig. S1a, b) and are relatively less frequent after
1980 for ERA5 except for 1992-1995, 1998, and 2015 when strong El Niño (1992, 1998, 2015) or La Niña (1995) events or a
volcanic eruption (1992) occurred (Fig. S1c). It reveals that the detected change points during 1990-2021 in ERA5 might be
associated with the result of abrupt climate changes. Previous studies also demonstrated the reliability of ERA5 air temperature
for the warming trend of the global ocean (He et al., 2023; Wang et al., 2019). For TPW, discontinuities show a similar pattern
as the temperature in ERA5, in which water vapor before 1980 has relatively more discontinuities. In addition, change points
are fewer after 1980 except for the largest discontinuity in 1991-1992 (Fig. S2a) in ERA5 and in 1996-1997 in JRA-55 (Fig.
S2b). Trenberth et al. (2015) highlighted that the discontinuity and inaccurate values in 1992 are the result of changes in
satellite instruments. A strong agreement between SSMI(S) observations and ERA5 is shown after 1993 (Allan et al., 2022),
and fewer discontinuities are presented from 1993 to 2021 (Fig. S2a). Meanwhile, discontinuities in JRA-55 are larger and
distributed in all decadal periods even in normal years (Fig. S2b, c). Because TPW is highly dependent on temperatures and
has a strong relationship with temperature over ocean, we further detected TPW discontinuities by comparing TPW and the
expected TPW (TPW$_{ept}$) calculated from regression with temperature (Figure S3) during 1981-2021 when temperature shows
fewer change points (Figure S1). The discrepancy points between TPW and TPW$_{ept}$ are obvious over 1987, 1998, 1999, and
2015 in both analysis data, indicating the strong effect of ENSO events on the change of TPW. For ERA5, these discontinuities
occurred in 1992 and 1995 when TPW$_{ept}$ increased but TPW decreased in ERA5 (Fig. S2), in which the discontinuity in 1992
also existed in ERA-I (Trenberth et al., 2015). For JRA-55, discrepancy points occurred in 1984 and 1995. Due to lack of
reliable observations before 1979, it is not recommended to use adjusted or statistically homogenized timeseries for trend
analysis without metadata confirmation (e.g., the homogenized observations) (Wang and Feng, 2013). Although several
discontinuities are detected in years after 1993 for TPW (Fig. S2), a strong agreement of TPW between ERA5 and SSMI
satellite might indicate reliable TPW trends (Allan et al., 2022). For the surface temperatures, most of the discontinuities occur
before the satellite era, therefore the long-term trend analysis for temperatures in reanalysis datasets after 1979 should be more
reliable, and removing the effect of discontinuities located in the pre-satellite period on longer-term trends needs further
investigation. In terms of temperature discontinuity identified (Figure S1), the longer temperature trends should be more
reliable by using surface observation networks' datasets (Pielke et al., 2007).

The RHARM radiosonde observations (Madonna et al., 2022), which are mostly located in the NH land and available from
1979-2019, are completely independent of reanalysis data in terms of calculating and adjusting temperature and humidity.
Compared with TPW trends from reanalysis data, moistening TPW in the NH polar region and Asia show the best agreement
with radiosondes, with more than 90% and 60% of stations' trend differences within ± 0.2 mm dec$^{-1}$, respectively (Fig. S6). In
addition, TPW trend differences are within ± 0.2 mm dec$^{-1}$ at more than 54% of stations in Europe and North America (Fig.
S6). Moistening TPW trends in Western Europe and South Asia and drying TPW trends in Western North America are
consistent with the results from both reanalysis data and GPS observations (Parracho et al., 2018). Overall, ERA5 and JRA-

55 show good agreement with observations at regional scales. Specifically, TPW shows better agreement in North America and the NH polar region.

After evaluating the reanalysis data with radiosonde observations on land, water vapor data from AIRS and SSMI(S) satellites are used to evaluate the accuracy of reanalysis over oceanic regions from 2003 to 2021 (Fig. S7). Except for the drying trends of the Northern Atlantic Ocean in the short-term period, the spatial drying trend patterns in the two reanalysis datasets are similar to the long-term trends (Fig.2) although they are not statistically significant. The surface air over tropical oceans increased in water vapor at a rate of 1.6 % dec$^{-1}$ for ERA5 and 1.9 % dec$^{-1}$ for JRA-55 on average, with a similar moistening rate shown in SSMI(S) (1.1% dec$^{-1}$) but not in AIRS data (0.05% dec$^{-1}$) since 2003. This result is consistent with Allan et al. (2022). The good agreement between the reanalyses and SSMI(S) is likely be attributed to the assimilation of all-sky radiances collected by the SSMI(S) satellite into the ERA5 and JRA-55 (Hersbach et al., 2020; Kobayashi et al., 2015). Although radiance measurement from AIRS is also introduced in ERA5 but not in JRA-55 (Hersbach et al., 2020; Kobayashi et al., 2015), the TPW from AIRS does not produce a strong global moistening and disagrees with reanalysis data and SSMI(S) microwave data and climate models, as well as Global Navigation System GPS (Allan et al., 2022; Douville et al., 2022).

**5 Conclusions**

Atmospheric reanalyses are widely used in the assessment of global climate change and their accuracy has advanced in recent years. Our study, bolstered by the latest ERA5 and JRA-55 reanalysis datasets, presents notable trends in the total precipitable water (TPW), surface temperature ($T_s$), and 2-meter temperature ($T_{2m}$) from 1993 to 2021. Beginning with TPW anomalies, a rise in moisture in response to global warming occurred from 1958-2021. However, due to data discontinuities in reanalysis data before 1993, more dependable trends have been identified. Post-1993, a moistening trend is observed at rates of 1.72% (ERA5) and 1.19% (JRA-55) per decade over oceans, 1.21% (ERA5) and 2.23% (JRA-55) per decade over land, and 2.12% (ERA5) and 1.90% (JRA-55) per decade globally. When considering regional TPW trends, we identified statistically significant moistening trends over North Africa, Europe, and Northern Hemisphere polar regions since 1993. These findings, which are in concert with prior studies (Borger et al. 2022; Trenberth et al. 2005; Wang et al. 2016), point to the solitary exception of the South Pacific Ocean, where TPW trends registered as negative. The observed warming pattern is unlike coupled simulation, with warming more prominent in the tropical warm pool, which can increase stability in these subtropical subsidence regions (Andrews et al., 2022). Reanalysis TPW showed an agreement with radiosonde observations over North America after 1979 and with SSMI(S) satellite but not AIRS measurement.

Building upon the analysis of TPW, our study further explored the temperature trends based on $T_{2m}$ and $T_s$ from ERA5 and JRA-55. Despite a general global warming trend (~ 0.15 K per decade), certain areas of the Southern Hemisphere and the North Atlantic Ocean demonstrated cooling since 1958. Terrestrial regions displayed a faster warming rate compared to

oceanic regions, with a ratio of roughly 2:1, corroborating findings from Swaminathan et al. (2022). Arctic warming was particularly pronounced, registering three times the global average during 1958-2021, and escalating to around four times from 1979 to 2021 and around 6.5% K$^{-1}$ of water vapor response to temperature. While Antarctic warming was more modest at 0.2 K per decade over the past 64 years, a sharp increase to over 0.6 K per decade was observed post-1980.

Last, we examined the TPW response to surface temperature changes, noting deviations from the Clausius-Clapeyron relation. In the ERA5 dataset, we identified TPW increases of around 11.2%, 5.8%, and 9.5% per K for ocean, land, and globe, respectively after 1993. These increased rates were higher in the JRA-55 dataset, with values at 10.9%, 9.4%, and 8.9% per K, respectively. Importantly, these response ratios were not uniform globally, presenting a variation between 6 and 8% K$^{-1}$ between latitudes 15 - 55°N and increasing toward southern high latitudes over oceans. The most substantial ratios for deviations in TPW responses were discovered in the southern high latitudes across land, ocean, and the globe. Because of the importance of atmospheric water vapor in the global energy balance and hydrological cycle, it has received much attention in recent years (this study; Trent et al., 2023; Douville and Willett, 2022; Patel and Kuttippurath, 2023; Shao et al., 2023; Ding et al., 2022). In the near future, results from these studies need to be synthesized to further quantify atmospheric water vapor, its relationship with surface temperature, and associated uncertainties. These results can then be used to better evaluate climate models and constrain these models' future projections.

*Code and data availability.* All data used for this study are freely available. The ERA5 dataset is openly available from ECMWF: https://cds.climate.copernicus.eu/cdsapp#!/dataset/reanalyses-era5-single-levels-monthly-means?tab=overview. The JRA-55 dataset is available from https://rda.ucar.edu/datasets/ds628.0/. The AIRS satellite data is from https://disc.gsfc.nasa.gov/datasets/AIRS3STD_006/summary. The SSMI(S) satellite is from www.remss.com/missions/ssmi. The Radiosounding HARMonization dataset is from https://cds.climate.copernicus.eu/cdsapp#!/dataset/insitu-observations-igra-baseline-network?tab=doc.

*Author contributions.* NW performed the analysis of the results and the visualization. XL was responsible for the funding acquisition. The original manuscript was written by NW and revised by XL, RPs, XZ, and AN.

*Competing interests.* The corresponding author has stated that there are no competing interests among the authors.

*Acknowledgments.* We thank Ryan Maue for inspiring this study as well as Dallas Staley and Stephen Watson for editing and finalizing the paper.

*Financial support.* This study was supported in part by the U.S. Department of Agriculture, Agricultural Research Service (A22-0103-001) and National Science Foundation (grant no. FAIN:2345039). The contribution number of this manuscript is 23-057-J.

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
