# Peer review of "Global total precipitable water variations and trends during 1958-2021"

_Hydrology and Earth System Sciences, 2023_

## Author Comment (AC2)

**Point-by-point Responses for Reviewer 1 for hess-2023-301**

We appreciate the insightful comments and suggestions from both reviewers. Following their guidance, we have made significant changes to our manuscript to enhance its clarity, accuracy, and robustness. A brief outline of the major revisions includes:

1. We refined our analysis of temperature and total precipitable vapor (TPW) discontinuities in reanalysis datasets. Instead of using regional averages, we applied the Penalized Maximal F (PMF) test to each grid over the ocean. Our analysis now encompasses the trends and variations in TPW and temperature for the periods 1958-2021 and 1993-2021, respectively. Additionally, we provide detailed trends, including the uncertainties at 95% confidence levels.

2. In response to Reviewer 2's suggestions, we revised our change point detection method, leading to the deletion of the original Tables S1 to S4 from the supplement. Consequently, Figures 1 to 6 in the main text have been updated, and Figures S1 to S5 have been added to the supplemental material.

3. We have carefully incorporated all the references and citations recommended by the reviewers into our revised manuscript.

The authors have used regular fonts for the Referee's comments (which might be divided into two or multiple comments), blue fonts for our responses, and red fonts with quotation marks to show the revised text. The line number in Reply refers to the new revision without tracked changes.

**Reviewer 1 # Comments**

The authors provide a valuable update in the regional changes in atmospheric water vapour content and surface temperature since 1958 based on state of the art reanalysis systems. While earlier attempts to use reanalyses demonstrated serious defects in homogeneity over time, the newer products appear more reliable and the analysis presented is complimentary to other work. Nevertheless, as noted in the discussion, the early satellite era may contain spurious variability related to changes in the observing system assimilated into the reanalysis model while the early record may be closer to a dynamically nudged "amip" atmosphere-only climate model simulation with prescribed SST and sea ice, though radiosonde information should provide some observational input in well sampled locations. The present work provides valuable information for studies interested in evaluating climate model simulations of moist processes including water vapour feedback. There is also recent interest in discrepancies between simulations and observations of low level moisture over arid and semi arid regions that could be mentioned (Simpson et al. 2023 PNAS doi:10.1073/pnas.2302480120). Despite the improvement in water vapour changes, other aspects of hydrological cycle such as precipitation show spurious global variability sincec 1979 (Allan et al. 2020 NYAS doi:10.1111/nyas.14337) so I wonder if there is any insight into this based on the water vapour evaluation? Overall, this is a well written and presented analysis that I recommend to be published with only minor modifications. A list of suggestions and comments is provided below.

Response: Thank you, Dr. Richard Allan, for your thorough review. Your encouragement and insights are valuable and substantially improved the quality of our paper. We agree Simpson et al. 2023 PNAS is an excellent reference for arid and semi-arid regions. We added two sentences in our revision in L179 as follows:

"In addition, Simpson et al. (2023) found that near-surface water vapor, as measured by surface observations, hasn't increased over arid and semi-arid regions since 1980, a discrepancy compared to predicted results from simulations. These findings indicate there might be misrepresentations of

hydroclimate-related processes in simulations since climate models showed moistening trends associated with the increase in water vapor-holding capacity of a warmer atmosphere."

In terms of water vapor change in both near-surface and surface-to-top-of-atmosphere, we would think the discrepancy between them is complicated and dynamic (scales). However, findings from surface observations by Simpson et al. provide us with references for those trends in arid and semi-arid regions in the world. Therefore, we also calculated TPW trends from 1980 to 2020 in reanalysis and discussed with observation results in Simpson et al. 2023. We added sentences in L193 as follows:

"In terms of TPW trend over arid and semi-arid regions from 1980 to 2020 in reanalysis dataset (figure not shown), it also shows a different trend, comparing observation from Simpson et al. 2023, that only TPW over southwest United States in ERA5 and central African in JRA-55 significantly decreased."

1) L10 should this be water vapor responses to lower tropospheric temperature (the thermodynamic causal route)? Radiative cooling rates affected by water vapor will also feedback on temperature but I don't think this is meant here?

Response: Thanks, we rewrote the sentence in L9 as

"Global responses of the hydrological cycle to climate change have been widely studied but uncertainties still remain regarding water vapor responses to lower tropospheric temperature."

2) L11 "improved" is ambiguous and can be removed

Response: done

3) L13 is the trend for the whole period?

Response: No, the period for radiosonde is from 1979 to 2019. The periods for Atmospheric Infrared Sounder (AIRS), and Microwave Satellite (SSMI(S)) observations are from 2003 to 2021. We rewrote the sentence in L10

"Here, we investigate the trends in global total precipitable water (TPW) and surface temperature from 1958 to 2021 using ERA5 and JRA-55 reanalysis datasets and further validate these trends by using radiosonde data from 1979 to 2019, and Atmospheric Infrared Sounder (AIRS) and Microwave Satellite (SSMI(S)) observations from 2003 to 2021."

4) L26 O'Gorman & Muller could be referred to on dTPW/dT

Response: Yes, we added O'Gorman and Muller (2010) reference in L28.

5) L28 "strengthened greenhouse effect"; could mention that changes higher in the troposphere are more important for the feedback while changes at low levels are strongly linked with precipitation

Response: This clarity is insightful. We rewrote the sentence in L28 as

"…due to the greenhouse effect (Held and Soden, 2006, O'Gorman and Muller, 2010) particularly in upper troposphere whereas changes at lower levels are strongly linked with precipitation patterns, influencing the frequency and intensity of extreme weather events (Trenberth, 1998; Trenberth et al., 2003)."

6) L35 see also the GPS network eg Douville et al. 2022 reference

Response: We have included Douville et al. (2022) in L59 as

"… with better accuracy, and relatively fewer homogeneity issues as both assimilate a huge amount of conventional and satellite-based observations (Hersbach et al., 2020; Kobayashi et al., 2015; Douville et al., 2022)."

7) L45 although reanalyses are much improved over earlier versions there remain some homogeneity issues eg before the mid 1990s over the tropical ocean eg Allan et al. 2022.

Response: We agreed that there remain some homogeneity issues in reanalysis datasets and rewrote the sentence in L62

"Many studies have confirmed that ERA5 is the best or among highest-performing reanalysis products (Taszarek et al., 2021; Yuan et al., 2023) although inhomogeneity still remains; for example, water vapor associated with changes in SSMI instruments (Trenberth et al., 2015) and unreliability of tropical water vapor in ERA5 and ground-based observations before 1993 (Allan et al., 2022)."

8) L141 - consistent --> inconsistent since the ERA5 decline in TPW in the 1980s is not consistent with SMMR/SSM/I microwave satellite data and appears to originate in the tropical lower troposphere over the ocean.

Response: Thank you, we corrected it in L141:

"A notable decrease in TPW is shown during the 1980s to 1990s over tropical regions in ERA5, which is consistent with ERA-interim (Allan et al., 2014) but inconsistent with satellite microwave data (Allan et al., 2022). This discrepancy seems to originate in the tropical lower troposphere over ocean."

9) L150 can the turning points be linked to phases of Pacific Decadal "Oscillation"? The jumps in TPW also seem to coincide with rapid increases in global temperature.

Response: Thank you, the change of TPW can be explained by the interdecadal Pacific oscillation (IPO). Study reveal that the trend during 1988-2003 was likely associated with the decadal variation of IPO from a warm period (1977-1998) to a cold period (1999 to 2003) (Dong and Dai, 2015). The PW trend for recent period including 1997/98 event will likely be similar (Wang et al. 2016). In addition, Patel and Kuttippurath (2023) also shows a strong correlation (0.81) between PDO and TPW variability in the tropics. We added sentences in L164 as follows:

"The trend of TPW during 1988-2003 was likely associated with the decadal variation of Interdecadal Pacific Oscillation from a warm period (1977-1998) to a cold period (1999 to 2003) (Dong and Dai, 2015). The TPW trend for recent period including 1997/98 event will likely be similar (Wang et al. 2016). In addition, Patel and Kuttippurath (2023) also shows a strong correlation (0.81) between Pacific Decadal Oscillation and TPW variability in the tropics."

10) L167 the North Atlantic cooling seems to be most prominent in the northern hemisphere winter (Allan & Allan 2020 JGR doi:10.1029/2019JC015379 ) while it is more a warming "hole" in the summer. Some have linked this with changes in ocean circulation rather than heat fluxes though this is based more on modelling (e.g. Drijfhout et al. 2012 doi:10.1175/JCLI-D-12-00490.1).

Response: We rewrote the sentence in L208

"Meanwhile, the North Atlantic cooling appears most significant in February in the northern hemisphere while it becomes net warming in August (Allan and Allan, 2020). Some studies connect this phenomenon to changes in ocean circulation based on modeling results (Drijfhout et al., 2012). In addition, based on model simulations without considering variable ocean currents, He et al. (2022) explained that the warming hole is driven by enhanced surface westerly wind removing heat from ocean surface. "

11) L190 the rapid Arctic warming is consistent with large increases in water vapour that can be menioned here (also discussed briefly in Allan et al. 2022) or signpost to the next section.

Response: We added the following sentence in L236:

"The Arctic warming is concentrated in the lower troposphere (Allan et al. 2022), and reached ~0.74 K dec$^{-1}$ after 1993, thus leading to increases in water vapor as shown in Fig. 2a."

12) L195 the reduction in relative humidity related to land/sea warming contrast could also be mentioned (e.g. Byrne & O'Gorman 2019 PNAS doi:10.1073/pnas.1722312115).

Response: We rewrote the sentence in L246 as follows:

"Byrne & O'Gorman (2018) further extended Joshi's theory in a quantitative way. They investigated the contrast of land/ocean warming from surface observations and model simulations at the global scale and indicated that amplified land temperature increases were the consequence of reduced relative humidity over land."

13) L206 O'Gorman & Muller (2010) would also be appropriate to cite here

Response: We added this reference in the sentence:

"… is constant (Trenberth et al., 2005; O'Gorman and Muller, 2010)."

14) L210 some regions are likely to exhibit more realistic trends where there is a greater density and homogeneity of data (e.g. N America, Europe)

Response: We added the sentence in L268:

"Regions with higher data density and homogeneity, such as North America and Europe, are likely to exhibit more realistic trends."

15) L215 it could be noted that (e.g. wet part of circulation moves from one location to another). This could be investigated based on reanalysis vertical motion fields for example.

Response: We rewrote the sentence in L269:

"Negative dTPW/dT occurs over the southern tropical ocean, where sea surface temperature was increased (Fig. 3) but precipitable water appears to be decreased slightly (Fig. 1); thus, the TPW change is contrary to what might be expected from the thermodynamic response to changes in temperature. Water vapor moves from one location to another through circulation, and the trend of vertical motion at 500 hPa could be further investigated to reveal the ascent and convective activity as well as its relationship to decreasing water vapor (Zveryaev and Allan, 2005)."

16) Figure 5 colour bar is not very intuitive to me (red implies drier to me)?

Response: To be consistent with response to reviewer 2, we deleted the TPW response to surface temperature ($T_s$), and calculated the ratio using data from 1993 to 2021. Figure R1 now serves as new Figure 5 in revision.

[Figure]

**Figure R1. (a)** The ratio (% K$^{-1}$) of TPW trends and surface air temperature ($T_{2m}$) trends from 1993 to 2021 in ERA5. **(b)** Same as **(a)** but for JRA-55**.** The hatch areas represent ratios that are not statistically significant at the 95% confidence level.

17) L253 - I think the SSM/I satellites only began in 1987, well after 1977?

Response: Thank you for pointing out this. You are correct. In the original table, the row was shifted downward by mistake when filled. However, based on reviewer 2's suggestion for achieving a more reliable detection, the table was deleted in our revision. We reanalyzed the discontinuity of each grid over the ocean instead of a regional average. The results are shown in Figures R2 and R3 and along with the following text added in L311, and Figures R2 and R3 serve as Figs. S2 and S3 in our revision.

"In order to get a general picture of discontinuities in the reanalysis data, we calculated the discontinuity from 1958 to 2021 for each grid box of temperature and TPW over the ocean, where there is a strong relationship between water vapor and temperature along with significantly increased trends (Figs. 1 and 3). We applied the PMF test on each grid of the reanalysis dataset and selected the significance level of 0.01 to detect discontinuities (Zhou et al., 2021). The years with detected discontinuities (after counting all change points within the same year) of both temperature and TPW are shown in Figures S1 and S2. Discontinuities for temperature are relatively frequent before 1980 in both reanalysis datasets (Fig. S1a, b) and are relatively less frequent after 1980 for ERA5 except for 1992-1995, 1998, and 2015 when strong EI Niño (1992, 1998, 2015) or La Niña (1995) events or a volcanic eruption (1992) occurred (Fig. S1c). It reveals that the detected change points during 1990-2021 in ERA5 might be associated with the result of abrupt climate changes. Previous studies also demonstrated the reliability of ERA5 air temperature for the warming trend of the global ocean (He et al., 2023; Wang et al., 2019). For TPW, discontinuities show a similar pattern as the temperature in ERA5, in which water vapor before 1980 has relatively more discontinuities. In addition, change points are fewer after 1980 except for the largest discontinuity in 1991-1992 (Fig. S2a) in ERA5 and in 1996-1997 in JRA-55 (Fig. S2b). Trenberth et al. (2015) highlighted that the discontinuity and inaccurate values in 1992 are the result of changes in satellite instruments. A strong agreement between SSMI(S) observations and ERA5 is shown after 1993 (Allan et al., 2022), and fewer discontinuities are presented from 1993 to 2021 (Fig. S2a). Meanwhile, discontinuities in JRA-55 are larger and distributed in all decadal periods even in normal years (Fig. S2b, c). Because TPW is highly dependent on temperatures and has a strong relationship with temperature over ocean, we further detected TPW discontinuities by comparing TPW and the expected TPW (TPW$_{ept}$) calculated from regression with temperature (Figure S3) during 1981-2021 when temperature shows fewer change points (Figure S1). The discrepancy points between TPW and TPW$_{ept}$ are obvious over

1987, 1998, 1999, and 2015 in both analysis data, indicating the strong effect of ENSO events on the change of TPW. For ERA5, these discontinuities occurred in 1992 and 1995 when $TPW_{ept}$ increased but TPW decreased in ERA5 (Fig. S2), in which the discontinuity in 1992 also existed in ERA-I (Trenberth et al., 2015). For JRA-55, discrepancy points occurred in 1984 and 1995. Due to lack of reliable observations before 1979, it is not recommended to use adjusted or statistically homogenized timeseries for trend analysis without metadata confirmation (e.g., the homogenized observations) (Wang and Feng, 2013). Although several discontinuities are detected in years after 1993 for TPW (Fig. S2), a strong agreement of TPW between ERA5 and SSMI satellite might indicate reliable TPW trends (Allan et al., 2022)."

[Figure]

**Figure R2.** Detected change points for monthly temperature during 1958-2021 in (a) ERA5 and (b) JRA55. (c) The timeseries of Oceanic Niño Index (ONI) from 1958 to 2021. There is total 21,307 of grids for PMF-test over the ocean.

[Figure]

**Figure R3**. The same as Figure RS5 but for monthly total water precipitation.

18) 267 - can the homogeneity tests be used to assess uncertainty in the computed trends (above the structural differences between reanalyses)? Trends also should be reported with at least statistical error bars relating to the linear fit

Response: This is an excellent question. The homogeneity tests are to detect the positions and magnitudes of change points, which are relevant to uncertainties of computed trends, but it is difficult to use them to assess the uncertainty of the trends. We agreed that trends should be reported with statistical error bars at a certain confidence level (e.g. 95%).

19) L270 - I am surprised that the radiosonde data is not assimilated in reanalyses? Is there a reason for this?

Response: The Radiosounding HARMonization (RHARM) Data Set applies the new algorithm to the Integrated Global Radiosonde Archive (IGRA) data, while IGRA is assimilated into reanalyses (Durre et al., 2018). The phrase "independent of reanalysis data" here refers to the way they used to calculate and adjust the systematic effects on fields (temperature and humidity). We rewrote the sentences in L101 and L343 as follows:

"We also used *in-situ* observations from the Radiosounding HARMonization (RHARM) dataset (Madonna et al., 2022) which applied the new algorithm to the Global Climate Observing System Reference Upper-Air Network (GRUAN) data and used observation measurement instead of reanalysis data as a reference to calculate and adjust for systematic effects on temperature and humidity."

"The RHARM radiosonde observations (Madonna et al., 2022), which are mostly located in the NH and available from 1979-2019, are independent of reanalysis data in terms of calculating and adjusting temperature and humidity."

20) L274 - why are trends reported in mm/decade here but %/decade earlier?

Response: To be consistent, the unit of TPW trend should be in %/decade. However, as reviewer 2 suggested that the quantile-matching (QM) procedure adjustment method we used is more appropriate for individual radiosondes. Therefore, we deleted the adjustment of reananlysis and only discussed the discontinuities. Sentences are added in L247 as response to Q17.

21) L284 - presumably the long term (1958-present) trends are mostly determined by recent rapid warming and moistening since the 1980s? It could be made clear that the values quoted here are since 2003?

Response: Yes, the trends over the long term (1958-present) are likely primarily influenced by the rapid warming and increased moisture levels observed since the 1980s from temperature variability in Figures 1 & 4 in main text. The values quoted here are from 2003 and we rewrote the sentence in L356 as follows:

"The surface air over tropical oceans increased in water vapor at a rate of 1.6 % dec$^{-1}$ for ERA5 and 1.9 % dec$^{-1}$ for JRA-55 on average, with a similar moistening rate shown in SSMI(S) (1.1% dec$^{-1}$) but not in AIRS data (0.05% dec$^{-1}$) since 2003."

22) L297 - are the decreases in the subtropical ocean cumulus transition zones explained by shifts in large-scale atmospheric circulation or changes in stability? It is noteworthy that the observed warming pattern is unlike coupled simulations, with warming more in the tropical warm pool which can increase stability in these subtropical subsidence regimes e.g. Andrews et al. 2022 JGR doi:10.1029/2022JD036675

Response: Following this comment, we added one sentence in L370 as:

"… registered as negative. The observed warming pattern is unlike coupled simulation, with warming more prominent in the tropical warm pool, which can increase stability in these subtropical subsidence regions (Andrews et al., 2022). When considering TPW …"

23) L308 - the link between Arctic warming and moistening could be usefully mentioned here

Response: We rewrote the sentence in L380

"Arctic warming was particularly pronounced, registering three times the global average during 1958-2021, and escalating to around four times from 1979 to 2021 and around 6.5% $K^{-1}$ of water vapor response to temperature."

24) L315 - a line of wider implications of the conclusions and future work would be welcome. Note that an intercomparison of TPW datasets is underway by Trent et al. https://doi.org/10.5194/egusphere-2023-2808. Some additional references that could also be considered are listed below:
Douville & Willett (2023) Sci. Adv. https://doi.org/10.1126/sciadv.ade6253
Patel & Kuttippurath (2023) OLA Research https://doi.org/10.34133/olar.0015
Shao et al. (2023) ACP https://doi.org/10.5194/acp-23-14187-2023
Ding et al. (2022) LNEE https://doi.org/10.1007/978-981-19-2588-7_27

Response: We added a short new paragraph at the end of Conclusion section as

"Because of the importance of atmospheric water vapor in the global energy balance and hydrological cycle, it has received much attention in recent years (this study; Trent et al., 2023; Douville and Willett, 2023; Patel and Kuttippurath, 2023; Shao et al., 2023; Ding et al., 2022). In the near future, results from these studies need to be synthesized to further quantify atmospheric water vapor, its relationship with surface temperature, and associated uncertainties. These results can then be used to better evaluate climate models and constrain these models' future projection."

**References** below are citations we added in our revision except papers with ** that are only used in this point-by-point response:

Allan, D. and Allan, R. P.: Seasonal Changes in the North Atlantic Cold Anomaly: The influence of cold surface waters from coastal Greenland and warming trends associated with Variations in subarctic sea ice cover, Journal of Geophysical Research: Oceans, 124, 9040–9052, https://doi.org/10.1029/2019JC015379, 2019.

Allan, R. P., Liu, C., Zahn, M., Lavers, D. A., Koukouvagias, E., and Bodas-Salcedo, A.: physically consistent responses of the global atmospheric hydrological cycle in models and observations, Surveys in Geophysics, 35, 533–552, https://doi.org/10.1007/s10712-012-9213-z, 2014.

Andrews, T., Bodas-Salcedo, A., Gregory, J. M., Dong, Y., Armour, K. C., Paynter, D., Lin, P., Modak, A., Mauritsen, T., Cole, J. N. S., Medeiros, B., Benedict, J. J., Douville, H., Roehrig, R., Koshiro, T., Kawai,

H., Ogura, T., Dufresne, J.-L., Allan, R. P., and Liu, C.: On the effect of historical SST patterns on radiative feedback, Journal of Geophysical Research: Atmospheres, 127, e2022JD036675, https://doi.org/10.1029/2022JD036675, 2022.

Byrne, M. P. and O'Gorman, P. A.: Trends in continental temperature and humidity directly linked to ocean warming, Proceedings of the National Academy of Sciences, 115, 4863–4868, https://doi.org/10.1073/pnas.1722312115, 2018.

Ding, J., Chen, J., and Tang, W.: Increasing trend of precipitable water vapor in Antarctica and Greenland, in: China Satellite Navigation Conference (CSNC 2022) Proceedings, Singapore, 286–296, https://doi.org/10.1007/978-981-19-2588-7_27, 2022.

Dong, B. and Dai, A.: The influence of the Interdecadal Pacific Oscillation on temperature and precipitation over the Globe, Clim Dyn, 45, 2667–2681, https://doi.org/10.1007/s00382-015-2500-x, 2015

Douville, H. and Willett, K. M.: A drier than expected future, supported by near-surface relative humidity observations, Science Advances, 9, eade6253, https://doi.org/10.1126/sciadv.ade6253, 2023.

Drijfhout, S., Oldenborgh, G. J. van, and Cimatoribus, A.: Is a decline of AMOC causing the warming hole above the North Atlantic in observed and modeled warming patterns?, Journal of Climate, 25, 8373–8379, https://doi.org/10.1175/JCLI-D-12-00490.1, 2012.

**Durre, I., Yin, X., Vose, R. S., Applequist, S., and Arnfield, J.: Enhancing the data coverage in the integrated global radiosonde archive, Journal of Atmospheric and Oceanic Technology, 35, 1753–1770, https://doi.org/10.1175/JTECH-D-17-0223.1, 2018.

He, C., Clement, A. C., Cane, M. A., Murphy, L. N., Klavans, J. M., and Fenske, T. M.: A North Atlantic warming hole without ocean circulation, Geophysical Research Letters, 49, e2022GL100420, https://doi.org/10.1029/2022GL100420, 2022.

He, M., Qin, J., Lu, N., and Yao, L.: Assessment of ERA5 near-surface air temperatures over global Oceans by Combining MODIS Sea Surface Temperature Products and In-Situ Observations, IEEE Journal of Selected Topics in Applied Earth Observations and Remote Sensing, 16, 8442–8455, https://doi.org/10.1109/JSTARS.2023.3312810, 2023.

Patel, V. K. and Kuttippurath, J.: Increase in tropospheric water vapor amplifies global warming and climate change, Ocean-Land-Atmosphere Research, 2, 0015, https://doi.org/10.34133/olar.0015, 2023.

Shao, X., Ho, S.-P., Jing, X., Zhou, X., Chen, Y., Liu, T.-C., Zhang, B., and Dong, J.: Characterizing the tropospheric water vapor spatial variation and trend using 2007–2018 COSMIC radio occultation and ECMWF reanalysis data, Atmospheric Chemistry and Physics, 23, 14187–14218, https://doi.org/10.5194/acp-23-14187-2023, 2023.

Simpson, I. R., McKinnon, K. A., Kennedy, D., Lawrence, D. M., Lehner, F., and Seager, R.: Observed humidity trends in dry regions contradict climate models, Proceedings of the National Academy of Sciences, 121, e2302480120, https://doi.org/10.1073/pnas.2302480120, 2023.

Trenberth, K. E.: Atmospheric moisture residence times and cycling: implications for rainfall rates and climate change, Climatic Change, 39, 667–694, https://doi.org/10.1023/A:1005319109110, 1998.

Trenberth, K. E., Dai, A., Rasmussen, R. M., and Parsons, D. B.: The changing character of precipitation, Bulletin of the American Meteorological Society, 84, 1205–1218, https://doi.org/10.1175/BAMS-84-9-1205, 2003.

Trenberth, K. E., Zhang, Y., Fasullo, J. T., and Taguchi, S.: Climate variability and relationships between top-of-atmosphere radiation and temperatures on Earth, Journal of Geophysical Research: Atmospheres, 120, 3642–3659, https://doi.org/10.1002/2014JD022887, 2015.

Trent, T., Schroeder, M., Ho, S.-P., Beirle, S., Bennartz, R., Borbas, E., Borger, C., Brogniez, H., Calbet, X., Castelli, E., Compo, G. P., Ebisuzaki, W., Falk, U., Fell, F., Forsythe, J., Hersbach, H., Kachi, M., Kobayashi, S., Kursinsk, R. E., Loyola, D., Luo, Z., Nielsen, J. K., Papandrea, E., Picon, L., Preusker, R., Reale, A., Shi, L., Slivinski, L., Teixeira, J., Vonder Haar, T., and Wagner, T.: Evaluation of total column water vapour products from satellite observations and reanalyses within the GEWEX water vapor Assessment, Climate and Earth System/Remote Sensing/Troposphere/Physics (physical properties and processes), https://doi.org/10.5194/egusphere-2023-2808, 2023.

Wang, C., Graham, R. M., Wang, K., Gerland, S., and Granskog, M. A.: Comparison of ERA5 and ERA-Interim near-surface air temperature, snowfall and precipitation over Arctic sea ice: effects on sea ice thermodynamics and evolution, The Cryosphere, 13, 1661–1679, https://doi.org/10.5194/tc-13-1661-2019, 2019.

Zhou, C., Wang, J., Dai, A., and Thorne, P. W.: A new approach to homogenize global subdaily Radiosonde Temperature Data from 1958 to 2018, Journal of Climate, 34, 1163–1183, https://doi.org/10.1175/JCLI-D-20-0352.1, 2021.

Zveryaev, I. I. and Allan, R. P.: Water vapor variability in the tropics and its links to dynamics and precipitation, Journal of Geophysical Research: Atmospheres, 110, https://doi.org/10.1029/2005JD006033, 2005.

---

## Author Comment (AC3)

**Point-by-point Responses for Reviewer 2 for hess-2023-301**

We appreciate the insightful comments and suggestions from both reviewers. Following their guidance, we have made significant changes to our manuscript to enhance its clarity, accuracy, and robustness. A brief outline of the major revisions includes:

1. We refined our analysis of temperature and total precipitable vapor (TPW) discontinuities in reanalysis datasets. Instead of using regional averages, we applied the Penalized Maximal F (PMF) test to each grid over the ocean. Our analysis now encompasses the trends and variations in TPW and temperature for the periods 1958-2021 and 1993-2021, respectively. Additionally, we provide detailed trends, including the uncertainties at 95% confidence levels.

2. In response to Reviewer 2's suggestions, we revised our change point detection method, leading to the deletion of the original Tables S1 to S4 from the supplement. Consequently, Figures 1 to 6 in the main text have been updated, and Figures S1 to S5 have been added to the supplemental material.

3. We have carefully incorporated all the references and citations recommended by the reviewers into our revised manuscript.

The authors have used regular fonts for the Referee's comments (which might be divided into two or multiple comments), blue fonts for our responses, and red fonts with quotation marks to show the revised text. The line number in Reply refers to the new revision without tracked changes.

**Reviewer 2 # Comments**

1. The topic as highlighted by the title is an important one, but one would not know from this paper that a comprehensive analysis already occurred by Allan et al. 2022. Although the latter is referred to in a couple of spots, buried in the paper, readers would have no idea that a lot here is not new and what is new is likely suspect. From Allan et al abstract "Global-scale changes in water vapor and responses to surface temperature variability since 1979 are evaluated across a range of satellite and ground-based observations, a reanalysis (ERA5) and coupled and atmosphere-only CMIP6 climate model simulations. Global-mean column integrated water vapor increased by 1%/decade during 1988–2014 in observations and atmosphere-only simulations."

Response: Thank you for your thorough review. Your insights are valuable and enable us to improve our paper quality. Note that Dr. Allan is another reviewer (reviewer 1), suggesting moderate revisions.

A series of Dr. Allan's papers and your important papers during this review process are very helpful. We used both ERA5 and JRA-55 datasets in which ERA5 uses the ECMWF IFS model for data assimilation while JRA-55 uses the Japanese 55-year Reanalysis system. Both assimilate observational data from various sources including satellites, radiosondes, weather stations, and ocean buoys.

2. One should not use 1958 as a starting point for trends especially without showing time series first. Satellite data for many fields became available mainly from 1978 on, and that is highly relevant over oceans for temperatures, but for PW it was only after mid 1987 when SSM/I data began. Allan et al also made use of SSMR data for 1979-1984. ERA5 data are flawed prior to about 1992 when the volume of SSMI data increased substantially (see Allan et al Fig 1b). In general pw data are unreliable over the oceans prior to 1987. Earlier SSMR data were not used in ERA5; see Fig 5 of Hersbach et al. Even after 1987 changes in microwave instruments and orbits caused further discontinuities, notably in 1992 (Trenberth et al 2015). That was evident for ERA-interim but it also holds for ERA5 as can be seen in the

time series in Fig 2f of this paper for the oceans, (less so for JRA55).  However, this was missed by this paper.

Response: We agree that TPW is more reliable after 1992 and acknowledge the data quality concerns identified by Allan et al. (2022) and Hersback et al. (2020). Accordingly, we changed our results-reporting sequences in order to firstly present a TPW timeseries aligning with a discussion of long-term (1958-2021) and short-term (1993-2021) periods, respectively. Our revision now in L138 is as follows:

"Figures. 1a-e show the temporal variations of monthly TPW anomalies over tropical, temperate, and polar regions. Overall, in terms of latitude bands except for tropical regions, TPW variability differs little between the two reanalysis datasets. Specifically, ERA5 gives a 0.6 % dec$^{-1}$ moistening rate whereas the JRA-55 shows a 0.97 % dec$^{-1}$ moistening rate over the tropical regions for the whole period 1958-2021. A notable decrease in TPW is shown from the late 1980s to early 1990s over tropical regions in ERA5 which is consistent with ERA-interim (Allan et al., 2014) but inconsistent with satellite microwave data (Allan et al., 2022). This discrepancy seems to originate in the tropical lower troposphere over the ocean. Since SSM/I satellites became available and were used in ERA5 after 1987 (Hersbach et. al., 2020), there has been some spurious variability in TPW, notably in January 1992 (Trenberth et al., 2015). However, a strong agreement of TPW after 1993 is shown between SSM/I observations and ERA5 (Allan et al., 2022). In addition, data after 1993 show fewer change points based on PMF-test results on grid boxes over the ocean (Figure S2). Thus, the 1993-2021 time series of TPW benefiting from in-situ and satellite observations is expected to produce more reliable water vapor variability. Specifically, ERA5 gives a 1.45 % dec$^{-1}$ moistening rate whereas the JRA-55 shows a 1.43 % dec$^{-1}$ moistening rate over the tropical regions from 1993 to 2021. Trends range from 1.64 to 3.89 % dec$^{-1}$ over temperate and polar regions in NH and are about 1.56 (0.89) % dec$^{-1}$ over temperate but are not significant over polar in SH for ERA5 (JRA-55) during 1993-2021."

Then in our revision, we presented the spatial context in L175 as:

"Based on timeseries of TPW in Fig.1, Fig. 2 shows the decadal trends in TPW calculated from ERA5 and JRA-55 after 1993. In general, the two reanalysis datasets showed similar trend patterns of global TPW distribution, with upward trends as the dominant change of TPW, indicating a rise in moisture in response to global warming since the end of the twentieth century (Santer et al., 2006). TPW trends are largely positive and statistically significant over North Africa, Europe and the NH polar region. In addition, Simpson et al. (2023) found that near-surface water vapor, as measured by surface observations, hasn't increased over arid and semi-arid regions since 1980, a discrepancy compared to predicted results from simulations. These findings indicate there might be misrepresentations of hydroclimate-related processes in simulations since climate models showed moistening trends associated with the increase in water vapor-holding capacity of a warmer atmosphere."

Figure R1 now serves as the new Figure 1 in revision.

[Figure]

**Figure R1.** Time series of monthly TPW anomalies (mm) over **(a)** Northern Hemisphere (NH) polar, **(b)** NH temperate, **(c)** Tropical, **(d)** Southern Hemisphere (SH) temperate, **(e)** SH polar, **(f)** ocean area, **(g)** land area, and **(h)** the globe from ERA5 (blue) and JRA-55 (red) during the period 1958–2021. A 12-month running smoother was applied to each time series shown. The trends are shown for 1958-2021 (light color) and 1993-2021 (dark color) periods with 95% confidence intervals in brackets. The vertical line indicates 1993 year.

Figure R2 now serves as the new Figure 2 in revision.

[Figure]

**Figure R2.** Monthly total precipitable water (TPW) trend (% dec$^{-1}$) from 1993 to 2021 from **(a)** ERA5 and **(b)** JRA-55. The hatch areas represent trends that are not statistically significant at a 95% confidence level.

In addition, we added a sentence in L201 as:

"In contrast to TPW, temperature in ERA5 is relatively stable for the pre-satellite era and aligned with independent datasets, although some notable discrepancies exist in accurately representing specific regional variations (Hersbach et al. 2020)."

And temperature trends after 1993 in 238 have been revised as:

"The Arctic warming is concentrated in lower troposphere (Allan et al. 2022) and reached ~0.74 K dec[-1] after 1993, thus leading to more water vapor as shown in Fig. 1a. Antarctic warming is at a rate of 0.24 K dec[-1] over the past 64 years and consistent warming at 0.23 K dec[-1] after 1993."

Meanwhile, we reanalyzed TPW change response to temperatures from 1993 to 2021 and reworded L259-L269 as:

"This relationship is determined by the ratio (dTPW/dT) of TPW trends and temperature trends from 1993 to 2021 (Fig. 5). The dTPW/dT ratio shows a similar pattern but different magnitude between two reanalysis datasets. The TPW increases significantly around 11.2% (10.9%), 5.8% (9.4%), and 9.5% (8.9%) per K for ocean, land, and globe for ERA5 (JRA-55). These rates are larger than 6% per K at a global scale based on observational HadCRUT5 (Allan et al., 2022). The dTPW/dT ratio increases significantly around 6% per K over eastern North America and Europe for both ERA5 and JRA-55 (Fig. 5). However, the relationship between TPW patterns and rising surface temperatures does not always follow the Clausius-Clapeyron equation, especially over some areas of Asia, central Africa, Australia and South American where TPW response to rising surface temperatures is negative without significance (Fig. 5). Regions with higher data density and homogeneity, such as North America and Europe, are likely to exhibit more realistic trends."

We additionally adjusted our text in L281 and L284 as,

"In addition to the global patterns of spatial dTPW/dT ratios, the ratio's latitude dependency is an interesting subject (Fig. 6) between 55° S and 55° N."

"The dTPW/dT ratios are close to the Clausius-Clapeyron response curves in NH mid- to high-latitudes over the ocean, but larger than 6% per K in the SH (-40 to -20°N band) over the ocean."

Figure R3 now serves as the new Figure 5 in revision.

[Figure]

**Figure R3. (a)** The ratio (% K[-1]) of the TPW trends and the surface air temperature ($T_{2m}$) trends from 1993 to 2021 in ERA5. **(b)** Same as (a) but in JRA-55. The hatch areas represent ratios that are not statistically significant at the 95% confidence level.

Figure R4 now serves as the new Figure 6 in revision,

[Figure]

**Figure R4**. The meridional means of change of TPW response to $T_{2m}$ for ERA5 (blue) and JRA-55 (red) over **(a)** globe, **(b)** land, and **(c)** ocean base using data from 1993 to 2021. The dashed lines represent the theoretically expected Clausius-Clapeyron response based on the climatological zonal mean temperature from the trend analysis.

3. Rawinsonde data have major issues through changes in the instruments and these have been noted in Trenberth et al (2005) and Zhou et al (2021), in addition to the papers included by Dai et al (2011) and Wang et al (2016). The homogenized data of Zhou et al. (2023) exhibit more spatially coherent trends and temporally consistent variations than the raw data, and lack the spurious tropospheric cooling over North China and Mongolia seen in several reanalyses and raw datasets, including ERA5.

Response: Following your comments, we reworded the description and discussion reflected in radiosonde observations in L36-49 in the introduction section as:

"The former has difficulty in evaluating long-term trends due to discontinuities, insufficient spatial densities, and coverage (Dee et al., 2011). For example, radiosonde observations have the issue of spurious discontinuities or changes in variability due to shifts in instruments and observational methods (Dee et al., 2011; Trenberth et al., 2005). However, multiple observations have been homogenized and thoroughly quality-checked in recent years. For example, the ground-based measurements of Global Navigation Satellite Systems (GNSS) integrated water vapor from 1994-2021 (Bock, 2022) and provided evidence of global moistening (Douville et al., 2022). The in-situ observations from the Radiosounding HARMonization (RHARM) dataset (Madonna et al., 2022) enhance the spatial consistency of estimated trends and align more closely with a contemporary atmospheric reanalysis. In addition, the homogenized temperature data presented by Zhou et al. (2021) exhibited spatially consistent trends and temporally stable variations, and didn't show the erroneous tropospheric cooling observed in various reanalyses including ERA5 and raw datasets, across North China and Mongolia. This accuracy of temperature can help in increasing the confidence in tropospheric temperature and water vapor, as well as in enhancing the quality of atmospheric reanalysis products. Temperature and humidity extremes from HadISDH by Willett (2023) are also designed for long-term regional trends."

and the issues in reanalysis dataset L49-55 in the introduction:

"The latter reanalysis dataset has issues of data quality that suffered from biases and errors from assimilated data during the satellite era, causing concerns about their reliability for detecting climate

trends (Dai et al., 2011; Schröder et al., 2016; Trenberth et al., 2011). Trenberth et al. (2005) evaluated the performance of global reanalyses, satellite, and radiosonde datasets on TPW and found that the Special Sensor Microwave Imager Sounder (SSM/I) dataset provided by Remote Sensing Systems (RSS) is the only source with reliable means, variabilities, and trends for TPW over oceans after 1988. The discontinuity and inaccurate data in 1992 due to changes in satellite instruments was highlighted by Trenberth et al (2015) and this spurious variability in water vapor existed in reanalyses up to the present."

And in L63:

"Many studies have confirmed that ERA5 is the best or among highest-performing reanalysis products (Taszarek et al., 2021; Yuan et al., 2023) although inhomogeneity still remains; for example, water vapor associated with changes in SSMI instruments (Trenberth et al., 2015) and unreliability of tropical water vapor in ERA5 and ground-based observations before 1993 (Allan et al., 2022)."

4. The relationships between pw and temperature are strong over the oceans, and regression can better reveal the discontinuities in pw, see Trenberth et al 2011, 2015.

Response: We regressed pw and temperature over the ocean, plotted the variability of pw and expected pw from 1981 to 2021 in Figure R5. The results show several discontinuities in both reanalyses, and we added sentences in L327, Figure R5 serves as Fig. S3 in supplement in our revision.

[Figure]

**Figure R5**. The variability of TPW and expected TPW calculated from regression with temperatures for ERA5 (a) and JRA-55 (b). (c) The timeseries of Oceanic Niño Index (ONI) from 1981 to 2021. Discrepancies are marked by triangles.

"Because TPW is highly dependent on temperatures and has a strong relationship with temperature over ocean, we further detected TPW discontinuities by comparing TPW and the expected TPW (TPW$_{ept}$) calculated from regression with temperature (Figure S3) during 1981-2021 when temperature shows fewer change points (Figure S1). The discrepancy points between TPW and TPW$_{ept}$ are obvious over 1987, 1998, 1999, and 2015 in both analysis data, indicating the strong effect of ENSO events on the change of TPW. For ERA5, these discontinuities occurred in 1992 and 1995 when TPW$_{ept}$ increased but TPW decreased in ERA5 (Fig. S2), in which the discontinuity in 1992 also existed in ERA-I (Trenberth et al., 2015). For JRA-55, discrepancy points occurred in 1984 and 1995. Due to lack of reliable observations before 1979, it is not recommended to use adjusted or statistically homogenized timeseries for trend analysis without metadata confirmation (e.g., the homogenized observations) (Wang and Feng, 2013). Although several discontinuities are detected in years after 1993 for TPW (Fig. S2), a strong agreement of TPW between ERA5 and SSMI satellite might indicate reliable TPW trends (Allan et al., 2022)."

5. However, even temperature data are unreliable over the oceans prior to satellite data, mainly 1979 or so, as noted about line 263.   But this reviewer does not support the procedures used to remove apparent discontinuities.

Response: We added additional explanation of discontinuities prior to 1979. However, adjusting reananlyses dataset to remove discontinuities is out of the scope of study. We added sentences in L338

"….after 1979 should be more reliable,  and removing the effect of discontinuities located in the pre-satellite period on longer-term trends needs further investigation."

6. It is not clear what this paper offers that is new and trustworthy?  The paper needs major revisions and should focus more on the time series and their relationships to highlight the observational issues and discontinuities.   Then maybe some more useful trends will emerge over times for which they are credible.

Response: We revised the manuscript (including enhancing discontinuity detection methods) by addressing all specific concerns raised by both reviewers. Through the analysis of two state-of-the-art reanalyses along with additional analysis using radiosonde data as well as AIRS and SSMI(S) satellite measurements, this study helps quantify TPW variations and trends. We also added a short paragraph at the end of Conclusions to address the implication of this work:

"Because of the importance of atmospheric water vapor in the global energy balance and hydrological cycle, it has received much attention in recent years (this study; Trent et al., 2023; Douville and Willett, 2023; Patel and Kuttippurath, 2023; Shao et al., 2023; Ding et al., 2022). In the near future, results from these studies need to be synthesized to further quantify atmospheric water vapor, its relationship with surface temperature, and associated uncertainties. These results can then be used to better evaluate climate models and constrain these models' future projection."

**Other major comments**

**Introduction**

7. There is no adequate introduction to past relevant studies.  Rather than Held and Soden 2006, Trenberth (1998) and Trenberth et al (2003) are more appropriate references for the importance and role of increasing water vapor.  Trenberth et al (2005) highlighted the issues of trends in pw in earlier years and noted the values over the ocean have no basis prior to microwave measurements from satellites that began

in 1987.   Even then changes in satellite instruments have led to discontinuities and erroneous values such as the transition in 1992 (Trenberth et al. 2015) that continues in the reanalyses to this day.  The introduction is especially remiss in not recognizing relevant studies, including Trenberth et al (2005), Zhou et al. (2021) who detail sonde issues up to date, and Allan et al (2022).  The latter provides analyses of the vertical structure of water vapor changes as well as trends for periods where they are credible.  Willett (2023) may also be useful.

Response: Prior TPW studies have been progressively updated including your papers, Dr. Allan's papers (Reviewer 1), including a few on-going studies he suggested, and two of Dr. Ben Santer's papers (who commented on our paper online). We carefully read and digested all of them and accordingly cited them in our revision. For the Introduction section, we revised the sentences at L28 as:

"…due to the greenhouse effect (Held and Soden, 2006, O'Gorman and Muller, 2010) particularly in upper troposphere whereas changes at lower levels are strongly linked with precipitation patterns, influencing the frequency and intensity of extreme weather events (Trenberth et al., 1998, 2003)."

We followed Dr. Allan's suggestion to discuss the vertical structure of water vapor changes in L269

"Negative dTPW/dT occurs over the southern tropical ocean, where sea surface temperature was increased (Fig. 3) but precipitable water appears to be decreased slightly (Fig. 1); thus, the TPW change is contrary to what might be expected from the thermodynamic response to changes in temperature. Water vapor moves from one location to another through circulation, and the trend of vertical motion at 500 hPa could be further investigated to reveal the ascent and convective activity as well as its relationship to decreasing water vapor (Zveryaev and Allan, 2005)."

In order to better introduce the potential issues in observations and instrument changes in datasets, we reworded sentences in L34-L38 in revised Introduction section as reply to Comment# 3.

Yes, Willett (2023) is useful and we added it in our revision.

**Surface temperature**

8. Skin temperature: what is the value of Ts?  This is not observed except when cloud free and it should be closely related to T2m over the oceans?  In any case no observations are used, as both Ts are model variables (lines 69-74) and will critically depend on cloud which is generally poorly simulated.  The paper should first straighten out the relationship between Ts and T2m in the supplemental, then use just one of those to relate to pw.  What is the point of showing both?  I recommend deleting all the panels in the figures with Ts.

Response: Surface skin temperature ($T_s$) is the theoretical (or simulated) temperature that represents the temperature of the uppermost surface layer. $T_s$ in the reanalysis dataset is based on model simulations and satellite-derived $T_s$ that both are influenced by clouds.

The relationship between $T_s$ and $T_{2m}$ are different over land and ocean. Considering that the ocean is homogeneous over large scales while land surface is more complex and highly variable in space, the correlation between $T_s$ and $T_{2m}$ is high over most of the oceans,  and the $T_{2m}$ variability is mainly forced by SST (Feng et al., 2018). However, the correlation over land depends strongly on vegetation fraction and land cover fraction (Good et al., 2017).

We reworded the sentences about the relationship of $T_s$ and $T_{2m}$, and Figures R6 & R7 serve as Figs. S4 and S5 in the supplement.

"The comparison of long-term changes between $T_s$ and $T_{2m}$ in reanalyses over the ocean, land and globe as shown in Figure S4. $T_s$ and $T_{2m}$ show very similar interannual variations although some discrepancies exist in years, especially prior to the 1980s. The two are much closer thereafter when multiple satellite observations are assimilated in reanalyses, indicating the overall coupling between $T_s$ and $T_{2m}$. In addition, the correlation of $T_s$ and $T_{2m}$ is strong (>0.9) and the trends aren't significantly different (student t-test) over ocean, land, and globe regions. For the variability over different latitude bins in Fig. S5 the discrepancy is the largest over polar area, with values around 0.4 K in JRA-55 and 0.2 K in ERA5. The difference between $T_s$ and $T_{2m}$ decrease after 1979 as more observations become available. Meanwhile, the trends of $T_s$ and $T_{2m}$ for the same latitude region aren't significantly different."

[Figure]

**Figure R6.** The relationship of monthly $T_{2m}$ and $T_s$ over land, ocean, and global for EAR5 (first column) and JRA-55 (second column).

[Figure]

**Figure R7.** The same as Figure R6 but for (a) Northern Hemisphere (NH) polar, (b) NH temperate, (c) Tropical, (d) Southern Hemisphere (SH) temperate, and (e) SH polar.

All the panels in the Figures with $T_s$ are deleted. Figure R8 now serves as new Figure 3 and Figure R9 serve as new Figure 4 in revision.

[Figure]

**Figure R8.** Monthly $T_{2m}$ anomaly relative to 1958 to 2021 over **(a)** Northern Hemisphere **(NH)** polar, **(b)** NH temperate, **(c)** Tropical, **(d)** Southern Hemisphere **(SH)** temperate, **(e)** SH polar, **(f)** ocean, **(g)** land,

and **(h)** the globe from the ERA5 (blue) and JRA-55 (red) data sets during the period of 1958 to 2021. A 12-month running smoother was applied to all time series. The trends are shown for 1958-2021 (light color) and 1993-2021 (dark color) periods with 95% confidence intervals in brackets. The vertical line indicates 1993 year.

[Figure]

**Figure R9.** Trends of surface air temperature ($T_{2m}$) from 1958 to 2021 for ERA5 **(a)** and JRA-55 **(b)**. The differential trends between $T_s$ and $T_{2m}$ trends for ERA5 **(c)** and JRA-55 **(d).** The hatch areas indicate trends that are not statistically significant. Trend units: K dec$^{-1}$.

**Issues with Discontinuities.**

9. The surface temperature record is well studied and there are several versions. Differences are mostly not large. But pw is fraught with huge sensitivity to changes in satellites and instruments, and basically few useful observations prior to microwave over the oceans in 1987. Even then spurious changes are readily evident. Some of these are discussed belatedly in the supplemental material. Evidently some adjustments are made, which is quite hazardous when one subsequently computes linear trends.

Response: Thanks for your insight, and we made revisions accordingly (see our reply to Comment #7). In addition, to achieve more reliable results, we first analyzed the discontinuities of TPW during 1958-2021 using PMF-test for each grid (see reply to Comment #10), then reanalyzed the trends of TPW from 1993 to 2021 and the relationship with temperature.

10. The discussion is lines 110 to 120. It seems the discontinuities were identified statistically, and the study did not use multiple regression (i.e. use temperature and pw together). Nor did it adequately deal with known issues in changes in satellites and their instruments. They did "defined the temporal coincidence when the discontinuity point positions were within a 3-month window along with observing measurement systems". "…the times series were adjusted using the quantile-matching (QM) procedure". Individual radiosondes might be corrected this way but those are likely readily dealt with by bias corrections in the data assimilation system.

I am quite bothered by these procedures, and I have no confidence they work. It would help to illustrate results in the supplemental. The suspicion is that they "corrected" some things that were not warranted and missed others. Take note of Zhou et al (2021) for example.

Response: The adjustment process in SI is a strict statistical method that has widely been used (and accepted) in surface climate change data and their homogeneity tests (e.g. Tom Karl and Tom Peterson in 1990s and 2000s). In surface climate data, we also used a multiple regression method to study multiple change points in one time series of one climate variable (e.g. surface air temperatures in surface climate observations). The method developed by Zhou et al (2021) is very similar to the method we used in SI. To our best knowledge, we think there are no fundamental differences in a statistical context.

Following this comment, we added Figures R10 and R11 and text in L311; Figures R10 and R11 serve as Figs. S1 and S2 in our revision.

"In order to get a general picture of discontinuities in the reanalysis data, we calculated the discontinuity from 1958 to 2021 for each grid box of temperature and TPW over the ocean, where there is a strong relationship between water vapor and temperature along with significantly increased trends (Figs. 1 and 3). We applied the PMF test on each grid of the reanalysis dataset and selected the significance level of 0.01 to detect discontinuities (Zhou et al., 2021). The years with detected discontinuities (after counting all change points within the same year) of both temperature and TPW are shown in Figures S1 and S2. Discontinuities for temperature are relatively frequent before 1980 in both reanalysis datasets (Fig. S1a, b) and are relatively less frequent after 1980 for ERA5 except for 1992-1995, 1998, and 2015 when strong El Niño (1992, 1998, 2015) or La Niña (1995) events or a volcanic eruption (1992) occurred (Fig. S1c). It reveals that the detected change points during 1990-2021 in ERA5 might be associated with the result of abrupt climate changes. Previous studies also demonstrated the reliability of ERA5 air temperature for the warming trend of the global ocean (He et al., 2023; Wang et al., 2019). For TPW, discontinuities show a similar pattern as the temperature in ERA5, in which water vapor before 1980 has relatively more discontinuities. In addition, change points are fewer after 1980 except for the largest discontinuity in 1991-1992 (Fig. S2a) in ERA5 and in 1996-1997 in JRA-55 (Fig. S2b). Trenberth et al. (2015) highlighted that the discontinuity and inaccurate values in 1992 are the result of changes in satellite instruments. A strong agreement between SSMI(S) observations and ERA5 is shown after 1993 (Allan et al., 2022), and fewer discontinuities are presented from 1993 to 2021 (Fig. S2a). Meanwhile, discontinuities in JRA-55 are larger and distributed in all decadal periods even in normal years (Fig. S2b, c)."

[Figure]

**Figure R10**. Detected change points for monthly temperature during 1958-2021 in (a) ERA5 and (b) JRA-55. (c) The timeseries of Oceanic Niño Index (ONI) from 1958 to 2021. There is a total of 21,307 grids for PMF-test over the ocean.

[Figure]

**Figure R11**. The same as Figure R10 but for monthly total water precipitation.

11. The issue is when the whole field is off because of a satellite change. An example is the spurious drop in pw values over the ocean in January 1992 when SSM/I changed (Trenberth et al 2015) that is not detected here, and this affects trends. Values after then should probably be relatively higher. This could be properly detected using a regression with temperatures for expected pw. i.e. assume the T record is adequate and predict the pw, then examine the discrepancies to better determine the discontinuities. In this way ENSO effects may also be included.

Response: Thank you for insights and suggestion. We analyzed the discrepancies between TPW and expected TPW as the reply for Comment#4. We would like to explore this approach in our on-going study. ENSO effects on change point's magnitudes and positions are intriguing because we agree ENSO effects might indeed impact them.

12. It is likely that there are no reliable trends prior to about 1992 and the maps presented are not regarded as useful. More reliable results are in Zhou et al (2021) and Allan et al (2022).

Response: We applied the PMF test on timeseries of each grid instead of regional averaged TPW and temperature as the reply for Comment #2. The conclusion confirms that the reliability of TPW after 1993, the map of the TPW and temperature relationship from 1993 to 2021 is updated thereafter in Figure R3.

13. In the supplemental material, Table S1 it says for ERA5 ocean "changing assimilation data from DMSP14 SSMI to 15" as 1977-01, but this occurred in 2000, not 1977.

Response: Thank you for pointing out this. In the original table, the rows were shifted downward by mistake when filled. However, as we replied to Comment #10, the PMF test is applied on each grid over the ocean and discontinuities of reanalyses are discussed.

**Some detailed comments**

L 63: this (discontinuities) should not be "lastly"

Response: Done.

L 74 regridded

Response: We changed it to "re-sampled"

L 89: why start the analysis in 1958 when there are no data over the oceans.

Response: We have now analyzed the dataset for full-time coverage and discussed the variability of long-term (1958-2021) and short-term (1993-2021) data, respectively.

L118: this adjustment procedure is not valid if one then computes trends. The whole treatment of inhomogeneities is cavalier, given the known problems documented in the literature.

Response: Following this comment, we decided not to use the quantile-matching (QM) procedure and only discussed the discontinuity using the PMF test (as the reply to Comment #10) and deleted L113-120 as well as tables S1-S4.

L 123: see discon discussion

Response: We rearranged these results-reporting, as the reply to Comment #2.

L 143: finally Allan et al 2022 is acknowledged. What is new here?

Response: We used different datasets from Dr. Allan and we have varied lengths although they have change points due to instrument changes. Our results supported Allan et al. 2002 findings. In addition, as Reviewer #1, Dr. Allan's first sentence is: "The authors provide a valuable update in the regional changes in atmospheric water vapour content and surface temperature since 1958 based on state of the art reanalysis systems."

L 165 why wouldn't they show remarkable similarities?

Response: This is part of the reason that we analyzed and presented both $T_s$ and $T_{2m}$. As we mentioned before, we used the student t- test to confirm that there is no significant difference between $T_{2m}$ and $T_s$ trends on a global scale.

L 179 Fig 3. Either panels a and d or b and e should be dropped, preferably include T2m. Also Fig 4 delete f,g,h,I,jn,o,p panels. Trends in surface temperature have appeared in many places and should not be the focus of this paper.

Response: Accordingly, we deleted all panels in the figures with $T_s$ and updated figures shown as Figures R3 to R6. Yes, our focus is the TPW in this paper.

L 195-198: The main reason for land ocean T change differences is surely that the land involves very shallow layers and the heat capacity differences are huge.

Response: We rewrote the sentence in L246 and added a new citation here (also based on Dr. Allan's review)

"Byrne & O'Gorman (2018) further quantitatively extended Joshi's theory. They investigated the contrast of land-ocean warming from surface observations and model simulations at the global scale and indicated that amplified land temperature increases were the consequence of the reduction of relative humidity over land."

L 215: whoa, Fig 5 has ugly colors, and the hatching makes it hard to decipher. Again why include 5b and 5d? This relationship over the ocean could be used to ferret out discontinuity issues and improve the reliability of results.

Response: Reviewer #1 also suggested us to change the colors for this figure. We deleted panels with $T_s$ and updated Fig. 5 (Figure R3 above) in our revision accordingly.

L 225: these results are likely wrong because of the failure to properly treat the discon issues.

Response: We updated Fig.5 by using data from 1993 to 2021, and reworded L285 and L294.

"The dTPW/dT ratios are close to the Clausius-Clapeyron response curves in NH mid- to high-latitudes over the ocean, but larger than 6% per K in the SH (-40 to -20ºN band) for ocean scales."

".. For land areas, in addition to two stronger response zones similar to the globe and ocean, the local maximum was found in the sub-tropical areas of both the NH and SH."

L 229: Fasullo (2011) is highly relevant here:

In deserts and arid regions, no amount of warming is matched by pw increases owing to absence of water. On the other hand, in monsoon regions moisture increases are amplified by atmospheric dynamics.

In our recent paper Cheng et al 2024 there is an analysis of the salinity that continues to show the fresh areas getting fresher and salty areas getting saltier: i.e. over the ocean, the wet areas are wetter and the dry areas are drier, and this is without complications of dry land. So shouldn't this also be expected this over land, subject to moisture availability?

Response: Following this comment, we added in L287

"In arid areas, due to a lack of water, warming doesn't result in increased water vapor; conversely, in monsoon regions, the dynamics of the atmosphere amplify moisture increases (Fasullo, 2011). Multiple other studies also confirm the "dry gets drier, and wet gets wetter" paradigm over land (Xiong et al., 2022). Likewise, an analysis of the salinity index in Cheng et al. (2024) showed that salty areas are

getting saltier and fresh areas are getting fresher; and that over the ocean wet areas are getting wetter and dry areas are getting drier."

L 241. This discussion should really have occurred earlier in the methods.

Response: Thank you. It is a logical suggestion. In our revision, we analyzed the timeseries of long-term TPW at first. Then, based on references (Trenberth et al. 2015, Allan et al. 2022), a short-term (after 1993) TPW was further analysed. The discontinuities are further discussed and confirm the reliability of TPW after 1993. Therefore, we would like to retain this sequence and the discontinuity issues are critical but we would like to keep our presentation sequence in revision.

L 291 on. It is unfortunate that the trends presented are for the entire period for which they are not credible. They would be much more useful if given for when the data are adequate to support the results.

Response: We followed your suggestion and changed this length of entire period into the period of 1993 to 2021.

References

Allan, R. P., Willett, K. M., John, V. O., and Trent, T.: Global Changes in Water Vapor 1979–2020, J. Geophys. Res., 127, e2022JD036728, https://doi.org/10.1029/2022JD036728, 2022.Cheng, L. J., J.

Abraham, K. E. Trenberth, T. Boyer, M. E. Mann, J. Zhu, F. Wang, F. J. Yu, R. Locarnini, J. Fasullo, F. Zheng, Y. L. Li, B. Zhang, L. Y. Wan, X. R. Chen, D. K. Wang, L. C. Feng, X. Z. Song, Y. L. Liu, F. Reseghetti, S. Simoncelli, V. Gouretski, G. Chen, A. Mishonov, J. Reagan, K. Von Schuckmann, Y. Y. Pan, Z. T. Tan, Y. J. Zhu, W. X. Wei, G. C. Li, Q. P. Ren, L. J. Cao, and Y. Y. Lu, 2024: New record ocean temperatures and related climate indicators in 2023, *Adv. Atmos. Sci.,* https://doi.org/10.1007/s00376-024-3378-5.

Fasullo J., 2011. A mechanism for land–ocean contrasts in global monsoon trends in a warming climate. *Clim Dyn*. DOI 10.1007/s00382-011-1270-3

Trenberth, K. E., 1998: Atmospheric moisture residence times and cycling: Implications for rainfall rates with climate change. *Climatic Change*, **39**, 667–694.

Trenberth, K. E., A. Dai, R. M. Rasmussen and D. B. Parsons, 2003: The changing character of precipitation. *Bull. Amer. Meteor. Soc.*, **84**, 1205-1217. https://doi.org/10.1175/BAMS-84-9-1205

Trenberth, K. E., J. Fasullo, and L. Smith, 2005: Trends and variability in column-integrated water vapor. *Clim. Dyn.,* **24***, 741-758

Trenberth, K. E., Y. Zhang, J. T. Fasullo, and S. Taguchi, 2015: Climate variability and relationships between top-of-atmosphere radiation and temperatures on Earth. *J. Geophys. Res.,* **120**, 3642-3659, https://doi.org/10.1002/2014JD022887

Willett, K. M. 2023: HadISDH .extremes Part I: A Gridded Wet Bulb Temperature Extremes Index Product for Climate Monitoring, *Adv. Atmos. Sci.,* doi: 10.1007/s00376-023-2347-8.

Zhou, C., Wang, J., Dai, A., & Thorne, P. W. (2021). A new approach to homogenize global subdaily radiosonde temperature data from 1958 to 2018. J.Climate, **34**, 1163–1183. https://doi.org/10.1175/JCLI-D-20-0352.1

Thank you for providing above references, which are very constructive and informative.

**References** below are citations we added in our revision except papers with ** that are only used in this point-by-point response:

Bock, O. Global GNSS Integrated Water Vapour data, 1994-2021 [Data set]. AERIS. https://en.aeris-data.fr/landing-page/?uuid=df7cf172-31fb-4d17-8f00-1a9293eb3b95

Allan, R. P., Liu, C., Zahn, M., Lavers, D. A., Koukouvagias, E., and Bodas-Salcedo, A.: physically consistent responses of the global atmospheric hydrological cycle in models and observations, Surveys in Geophysics, 35, 533–552, https://doi.org/10.1007/s10712-012-9213-z, 2014.

Cheng, L., Abraham, J., Trenberth, K. E., Boyer, T., Mann, M. E., Zhu, J., Wang, F., Yu, F., Locarnini, R., Fasullo, J., Zheng, F., Li, Y., Zhang, B., Wan, L., Chen, X., Wang, D., Feng, L., Song, X., Liu, Y., Reseghetti, F., Simoncelli, S., Gouretski, V., Chen, G., Mishonov, A., Reagan, J., Von Schuckmann, K., Pan, Y., Tan, Z., Zhu, Y., Wei, W., Li, G., Ren, Q., Cao, L., and Lu, Y.: New record ocean temperatures and related climate indicators in 2023, Advances in Atmospheric Sciences, https://doi.org/10.1007/s00376-024-3378-5, 2024.

Ding, J., Chen, J., and Tang, W.: Increasing trend of precipitable water vapor in Antarctica and Greenland, in: China Satellite Navigation Conference (CSNC 2022) Proceedings, Singapore, 286–296, https://doi.org/10.1007/978-981-19-2588-7_27, 2022.

Douville, H. and Willett, K. M.: A drier than expected future, supported by near-surface relative humidity observations, Science Advances, 9, eade6253, https://doi.org/10.1126/sciadv.ade6253, 2022.

Douville, H., Qasmi, S., Ribes, A., and Bock, O.: Global warming at near-constant tropospheric relative humidity is supported by observations, Communications Earth & Environment, 3, 1–7, https://doi.org/10.1038/s43247-022-00561-z, 2022.

Fasullo, J.: A mechanism for land–ocean contrasts in global monsoon trends in a warming climate, Climate Dynamics, 39, 1137–1147, https://doi.org/10.1007/s00382-011-1270-3, 2012.Feng, X., Haines, K., and de Boisséson, E.: Coupling of surface air and sea surface temperatures in the CERA-20C reanalysis, Quarterly Journal of the Royal Meteorological Society, 144, 195–207, https://doi.org/10.1002/qj.3194, 2018.

**Feng, X., Haines, K., and de Boisséson, E.: Coupling of surface air and sea surface temperatures in the CERA-20C reanalysis, Quarterly Journal of the Royal Meteorological Society, 144, 195–207, https://doi.org/10.1002/qj.3194, 2018.

**Good, E. J., Ghent, D. J., Bulgin, C. E., and Remedios, J. J.: A spatiotemporal analysis of the relationship between near-surface air temperature and satellite land surface temperatures using 17 years of data from the ATSR series, Journal of Geophysical Research: Atmospheres, 122, 9185–9210, https://doi.org/10.1002/2017JD026880, 2017.

He, M., Qin, J., Lu, N., and Yao, L.: Assessment of ERA5 near-surface air temperatures over global oceans by combining MODIS sea surface temperature products and in-situ observations, IEEE Journal of Selected Topics in Applied Earth Observations and Remote Sensing, 16, 8442–8455, https://doi.org/10.1109/JSTARS.2023.3312810, 2023.

Patel, V. K. and Kuttippurath, J.: Increase in tropospheric water vapor amplifies global warming and climate change, Ocean-Land-Atmosphere Research, 2, 0015, https://doi.org/10.34133/olar.0015, 2023.

Shao, X., Ho, S.-P., Jing, X., Zhou, X., Chen, Y., Liu, T.-C., Zhang, B., and Dong, J.: Characterizing the tropospheric water vapor spatial variation and trend using 2007–2018 COSMIC radio occultation and ECMWF reanalysis data, Atmospheric Chemistry and Physics, 23, 14187–14218, https://doi.org/10.5194/acp-23-14187-2023, 2023.

Simpson, I. R., McKinnon, K. A., Kennedy, D., Lawrence, D. M., Lehner, F., and Seager, R.: Observed humidity trends in dry regions contradict climate models, Proceedings of the National Academy of Sciences, 121, e2302480120, https://doi.org/10.1073/pnas.2302480120, 2023.

Trenberth, K. E., Zhang, Y., Fasullo, J. T., and Taguchi, S.: Climate variability and relationships between top-of-atmosphere radiation and temperatures on Earth, Journal of Geophysical Research: Atmospheres, 120, 3642–3659, https://doi.org/10.1002/2014JD022887, 2015.

Trent, T., Schroeder, M., Ho, S.-P., Beirle, S., Bennartz, R., Borbas, E., Borger, C., Brogniez, H., Calbet, X., Castelli, E., Compo, G. P., Ebisuzaki, W., Falk, U., Fell, F., Forsythe, J., Hersbach, H., Kachi, M., Kobayashi, S., Kursinsk, R. E., Loyola, D., Luo, Z., Nielsen, J. K., Papandrea, E., Picon, L., Preusker, R., Reale, A., Shi, L., Slivinski, L., Teixeira, J., Vonder Haar, T., and Wagner, T.: Evaluation of total column water vapour products from satellite observations and reanalyses within the GEWEX water vapor assessment, Climate and Earth System/Remote Sensing/Troposphere/Physics (physical properties and processes), https://doi.org/10.5194/egusphere-2023-2808, 2023.

Wang, C., Graham, R. M., Wang, K., Gerland, S., and Granskog, M. A.: Comparison of ERA5 and ERA-Interim near-surface air temperature, snowfall and precipitation over Arctic sea ice: effects on sea ice thermodynamics and evolution, The Cryosphere, 13, 1661–1679, https://doi.org/10.5194/tc-13-1661-2019, 2019.

Willett, K. M.: HadISDH.extremes Part I: A gridded wet bulb temperature extremes index product for climate monitoring, Advances in Atmospheric Sciences, 40, 1952–1967, https://doi.org/10.1007/s00376-023-2347-8, 2023.

Xiong, J., Guo, S., Abhishek, Chen, J., and Yin, J.: Global evaluation of the "dry gets drier, and wet gets wetter" paradigm from a terrestrial water storage change perspective, Hydrology and Earth System Sciences, 26, 6457–6476, https://doi.org/10.5194/hess-26-6457-2022, 2022.

Zhou, C., Wang, J., Dai, A., and Thorne, P. W.: A new approach to homogenize global subdaily radiosonde temperature data from 1958 to 2018, Journal of Climate, 34, 1163–1183, https://doi.org/10.1175/JCLI-D-20-0352.1, 2021.

Zveryaev, I. I. and Allan, R. P.: Water vapor variability in the tropics and its links to dynamics and precipitation, Journal of Geophysical Research: Atmospheres, 110, https://doi.org/10.1029/2005JD006033, 2005.